# Inverse Reinforcement Learning with Switching Rewards and History Dependency for Characterizing Animal Behaviors

## Abstract

Traditional approaches to studying decision-making in neuroscience focus on simplified behavioral tasks where animals perform repetitive, stereotyped actions to receive explicit rewards. While informative, these methods constrain our understanding of decision-making to short timescale behaviors driven by explicit goals. In natural environments, animals exhibit more complex, long-term behaviors driven by intrinsic motivations that are often unobservable. Recent works in time-varying inverse reinforcement learning (IRL) aim to capture shifting motivations in long-term, freely moving behaviors. However, a crucial challenge remains: animals make decisions based on their history, not just their current state. To address this, we introduce SWIRL (SWitching IRL), a novel framework that extends traditional IRL by incorporating time-varying, history-dependent reward functions. SWIRL models long behavioral sequences as transitions between short-term decision-making processes, each governed by a unique reward function. SWIRL incorporates biologically plausible history dependency to capture how past decisions and environmental contexts shape behavior, offering a more accurate description of animal decision-making. We apply SWIRL to simulated and real-world animal behavior datasets and show that it outperforms models lacking history dependency, both quantitatively and qualitatively. This work presents the first IRL model to incorporate history-dependent policies and rewards to advance our understanding of complex, naturalistic decision-making in animals.

## 1 Introduction

Historically, decision making in neuroscience has been studied using simplified assays where animals perform repetitive, stereotyped actions (such as licks, nose pokes, or lever presses) in response to sensory stimuli to obtain an explicit reward. While this approach has its advantages, it has limited our understanding of decision making to scenarios where animals are instructed to achieve an explicit goal over brief timescales, usually no more than tens of seconds. In contrast, in natural environments, animals exhibit much more complex behaviors that are not confined to structured, stereotyped trials. For example, a freely moving mouse may immediately rush toward the scent of food when hungry, but after eating, it might seek out a quiet spot to rest for an extended period. Thus, real-world animal behaviors form long sequences composed of multiple decision-making processes. Each decision-making process involves a series of states and actions aimed at achieving a goal, and such decision switching is unlikely to occur on very short timescales in simplified assays. Additionally, many of the goals animals pursue in natural settings are generated by intrinsic motivations and thus unobservable. To truly understand animal's decision-making in a naturalistic context, we need methods to uncover animals' intrinsic motivations during multiple decision-making processes.

Inverse reinforcement learning (IRL), which infers agents' policies and intrinsic reward functions based on their interactions with the environment (Ng & Russell, 2000; Abbeel & Ng, 2004; Ziebart et al., 2008; 2010; Wu et al., 2024), has been shown to be effective in capturing animal decision-making intentions by learning reward functions from behavioral trajectories (Sezener et al., 2014; Yamaguchi et al., 2018; Pinsler et al., 2018; Hirakawa et al., 2018). However, traditional IRL assumes a single static reward function over time, limiting its ability to account for shifts in intrinsic motivations. To address this limitation, recent IRL variants have aimed to uncover heterogeneous and time-varying reward functions (Babes-Vroman et al., 2011; Surana & Srivastava, 2014; Nguyen

et al., 2015; Ashwood et al., 2022a; Zhu et al., 2024). Despite these advancements, a significant challenge remains unaddressed: animals make decisions based on their history, not just their current state (Kennedy, 2022; Hattori et al., 2019). For example, in perceptual decision-making tasks, mice are found to make new decision based on reward, state and decision history (Ashwood et al., 2022b). Incorporating historical context into the modeling could offer a more accurate representation of animal behavior.

To address the absence of history dependency in time-varying IRL models, we introduce a novel framework called SWitching IRL (SWIRL). Similar to Zhu et al. (2024), SWIRL models long recordings of animal behaviors as a sequence of short-term decision-making processes. Each decision-making process is treated as a Markov decision process (MDP) with a unique reward function that can be inferred using IRL. The segmentation of a long recording into switching decision-making processes is unknown; therefore, each process is regarded as being associated with a hidden mode that must also be inferred. Most importantly, SWIRL incorporates biologically plausible history dependency, drawing on insights from animal behavior. The history dependency is added at two levels: the transitions between decision-making processes (decision-level) and the actions taken to achieve a single goal within a decision-making process (action-level). Decision-level dependency is reflected in the transitions between decision-making processes over extended sequences of time bins, suggesting that an animal's current choice is shaped by its previous decisions and environmental feedback. Additionally, we posit that these transitions are influenced by the animal's location. For example, after a mouse drinks from a water port, if it stays nearby, it is more likely to seek another goal. Conversely, if it is far from the port, indicating it has been away for some time, it may become thirsty again and return to search for water. For action-level history dependency, we will model the policy and reward functions as dependent on trajectory history within each decision-making process, using a non-Markovian decision framework. Such a dependency has been studied by existing reinforcement learning research, which often characterizes exploration with reward functions based on historical states and actions (Houthooft et al., 2016; Sharafeldin et al., 2024). Importantly, our paper is the first to incorporate history-dependent policies and rewards into IRL.

One key aspect we want to highlight in this paper is that the proposed SWIRL model has intriguing connections to traditional behavioral analysis methods in the animal neuroscience literature. In Sec. 3.5, we will demonstrate that our SWIRL model offers a more generalized and principled approach to characterize animal behaviors compared to existing autoregressive dynamics models (Wiltschko et al., 2015; Mazzucato, 2022; Stone, 2023; Weinreb et al., 2024).

In the Results section, we will apply our SWIRL to a simulated dataset as well as two real-world animal behavior datasets. For both animal datasets, we will demonstrate that SWIRL outperforms alternative models when history dependency is not included, both quantitatively and qualitatively. This underscores the necessity of incorporating this biologically plausible element when modeling long-term behaviors. Additionally, for the first time, we will present the application of non-Markovian reward functions and state-action reward functions to model freely-moving animals, contrasting with previous works that only assume a single state-based reward.

## 2 RELATED WORK

**IRL for animal behavior understanding.** IRL has been widely used to infer animals' behavioral strategies and decision-making policies when the reward is unknown. For instance, Pinsler et al. (2018) applies IRL to uncover the unknown reward functions of pigeons, explaining and reproducing their flock behavior, and developed a method to learn a leader-follower hierarchy. Similarly, Hirakawa et al. (2018) uses IRL to learn reward functions from animal trajectories, identifying environmental features preferred by shearwaters, and discovered differences in male and female migration route preferences based on the estimated rewards. In another study, Yamaguchi et al. (2018) apllies IRL to C. elegans thermotactic behavior, revealing distinct behavioral strategies for fed and unfed worms. Additionally, Sezener et al. (2014) maps reward functions for rats freely moving in a square area, showing how these rewards changed before and after training. While these studies demonstrate the utility of IRL in uncovering behavioral strategies of freely moving animals, they share a key limitation: they all assume a single reward function governs all animal behaviors, which does not account for the complexities of long-term decision-making.

**Heterogeneous and time-varying IRL.** Recent works have extended traditional IRL, which assumes a constant reward, to models with time-varying or multiple reward functions driving behav-

ioral trajectories. For example, Babes-Vroman et al. (2011) introduced Multi-intention IRL, which infers multiple reward functions across different trajectories but still assumes a single reward function within each trajectory. On the other hand, the Dynamic IRL (DIRL) method (Ashwood et al., 2022a) models reward functions as a linear combination of feature maps with time-varying weights, addressing the issue of varying rewards within a trajectory. However, DIRL requires trajectories to be highly similar or clustered beforehand, significantly limiting its applicability. Moreover, it cannot capture switching decision-making processes over long-term periods where each process may vary in length. Additionally, BNP-IRL (Surana & Srivastava, 2014), locally consistent IRL (Nguyen et al., 2015) and multi-intention inverse Q-learning (IQL) (Zhu et al., 2024) all extended the multi-intention IRL framework to allow for changing reward functions within trajectories, making them the closest models to our proposed SWIRL. However, all models do not account for both decision-level and action-level history dependency, an important biologically plausible factor that SWIRL incorporates to achieve more accurate behavior modeling. In our experiments, we will use multi-intention IQL and locally consistent IRL as baseline models, as they are special cases of SWIRL.

**Dynamics-based behavior analysis in animal neuroscience.** Traditional approaches to analyzing animal behavior in neuroscience often rely on autoregressive dynamics models. For instance, MoSeq and related works (Wiltschko et al., 2015; Weinreb et al., 2024) assume that animal behavior consists of multiple segments modeled by an HMM, with each segment evolving through an autoregressive process. Stone (2023) introduces a switching linear dynamical system (SLDS), similar to an AR-HMM, but with an additional layer of continuous latent states between the behavioral trajectories and the hidden states representing behavioral segments. We argue that if each segment lasts only a few seconds, it represents meaningful action motifs, such as grooming and sniffing. However, if a segment is significantly longer and reflects a decision-making process, traditional dynamics-based models may not be suitable for identifying these long-term segments. However, these dynamics-based models are not entirely independent of SWIRL. We will demonstrate that SWIRL generalizes purely dynamics-based models by relying on a more principled IRL framework to identify multiple decision-making processes. Our goal is to offer profound insights that bridge these traditional and new models for animal behavioral analysis.

## 3 METHODS

### 3.1 HIDDEN-MODE MARKOV DECISION PROCESS

A discounted Hidden-Mode Markov Decision Process (HM-MDP) is defined by the tuple $\mathcal{M} = (\mathcal{Z}, \mathcal{S}, \mathcal{A}, \mathcal{P}, \mathcal{P}_z, r, \gamma)$. Here, $\mathcal{Z}$ represents a finite set of hidden modes $z$, $\mathcal{S}$ denotes the finite state space, and $\mathcal{A}$ indicates the finite action space. $r_z$ represents the reward function $r$ under hidden mode $z$. The discount factor $\gamma$ is constrained to the interval $[0, 1]$. Starting from an initial state $s_0$, the agent (animal) selects an action $a$ based on its policy (behavioral strategy) $\pi$ and subsequently receives a reward determined by $r_z$, $z \in \mathcal{Z} := \{z_1, z_2, \ldots, z_m\}$, where $m$ represents the total number of modes. The agent then transitions to the next state $s'$ according to the transition kernel $\mathcal{P}(s'|s, a)$, while the agent's hidden mode also transitions to $z'$ based on the transition probability $\mathcal{P}_z(z'|z)$.

### 3.2 INVERSE REINFORCEMENT LEARNING

Inverse Reinforcement Learning (IRL) addresses the scenario where we have gathered multiple trajectories from an expert agent $\pi^*$, comprising a set of state-action pairs $\{(s_t^*, a_t^*)\}$. The goal is to estimate the policy and reward that generated these state-action pairs, often referred to as demonstrations in the literature. We assume that we have collected $N$ expert trajectories, denoted as $\mathcal{D} = \{\xi_1, \xi_2, \ldots, \xi_N\}$. Each trajectory consists of a sequence of state-action pairs, represented as $\xi_n = \{(s_1^*, a_1^*), (s_2^*, a_2^*), \ldots\}$, with $T_n$ time steps, which may vary across trajectories.

### 3.3 SWITCHING INVERSE REINFORCEMENT LEARNING

Our SWIRL model is built on the HM-MDP. Instead of explicitly knowing the reward for each mode, we will use IRL to infer these rewards. Mathematically, at each time step $t$, we represent the agent's internal reward function $r_{z_t}$ with an additional dependency on the hidden mode $z_t$. This means that the agent receives a reward $r_{z_t}$ based on its current hidden mode $z_t$, which indicates the decision-making state the animal is in (e.g., water seeking or home seeking), with $r_{z_t}$ representing the corresponding intrinsic motivation. Consequently, the optimal policy $\pi_t$ is determined by $r_{z_t}$. However, SWIRL goes beyond merely embedding IRL within HM-MDP. We also introduce two levels of history dependency into the model. The full graphical model is depicted in Fig. 1.

The decision-level dependency is characterized by the idea that animals make new decisions based on their previous choices. The transitions between decision-making processes already account for this decision-level dependency since $\mathcal{P}_z(z_{t+1}|z_t)$. However, the hidden modes with such a classical transition are generated through an open-loop process: the mode $z_{t+1}$ depends solely on the preceding mode $z_t$, with $z_{t+1}|z_t$ being independent of the observation state $s_t$. Consequently, if a discrete switch should occur when the animal enters a specific region of the state space, the classical transition will fail to capture this dependency. To address this, we extend the transition model to include the state $s_t$ as a condition, resulting in $\mathcal{P}_z(z_{t+1}|z_t, s_t)$, which effectively captures the desired relationship between decisions and the animal's location.

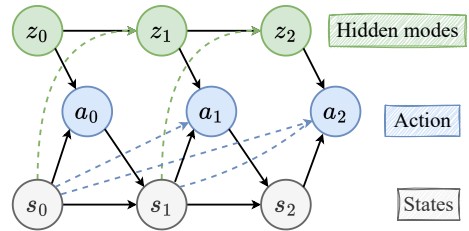

For action-level history dependency, we treat both the reward and policy under a hidden mode $z$ as functions dependent on the previous $L$ states, specifically $r_z : \mathcal{S}^L \times \mathcal{A} \to \mathbb{R}$ and $\pi_z : \mathcal{S}^L \to \mathcal{A}$, where $L \in \mathbb{N}$ and $\mathcal{S}^L$ denotes the cartesian product of $L$ state spaces. To simplify the notation, we denote an element of $\mathcal{S}^L$ as $s^L$, so that $r_z(s^L, a) := r_z(s^1, s^2, \ldots, s^L, a)$ and $\pi_z(a|s^L) := \pi_z(a|s^1, s^2, \ldots, s^L)$, and pad both functions with dummy variables if the current time step is less than $L$. We can also add the dependency of previous actions for the reward function. But we use state-only rewards for simplicity and the IRL tradition. It's straightforward to do so, though. This makes it natural to extend from a single state dependency to a history of state dependencies in our work.

Figure 1: SWIRL graphical model. Green dotted lines represent transitions of the hidden modes depend on the previous state (decision-level dependency). Blue dotted lines represent that polices depend on past states (action-level dependency).

Furthermore, we can view the decision process as being non-Markovian, meaning that the current decision or action depends not only on the current state but also on the history of previous states and actions. Noticeably, there are various approaches to address non-Markovian decision processes, including state augmentation (Sutton, 1991), recurrent neural networks (Bakker, 2001; Hausknecht & Stone, 2015), Neural Turing Machines (Parisotto & Salakhutdinov, 2017) and so forth. In this paper, we adopt the most common approach–state augmentation; however, the framework can also be implemented using more advanced and scalable methods.

### 3.4 SWIRL INFERENCE PROCEDURE

The goal of inference is to learn the hidden modes $z$ and the model parameters $\theta = (\mathcal{P}_z, r_z, \pi_z, p(s_1), p(z_1))$ given the collected trajectories $\mathcal{D}$. Here, $p(s_1)$ and $p(z_1)$ represent the probabilities of the initial state and hidden mode, respectively. The variables $r_z$ and $\pi_z$ denote the reward and policy associated with the hidden mode $z$, while $\mathcal{P}_z$ is the transition matrix between hidden modes. We can maximize the likelihood of the demonstration trajectories $\mathcal{D}$ to learn the optimal $\theta^*$, such that $\theta^* = \arg\max_\theta \log P(\mathcal{D}|\theta)$ (MLE). However, achieving this objective requires marginalizing over the hidden modes $z$, which is intractable. To address the intractability, we employ the Expectation-Maximization (EM) algorithm, alternating between updating the parameter estimates and inferring the posterior distributions of the hidden modes.

Following the EM update scheme, we derive the auxiliary function for the $n$-th trajectory during the E-step, where $n = 1, 2, \ldots, N$:

$$G_n(\theta, \hat{\theta}) = \log p(s_{n,1}) + \sum_z p(z_{n,1}|\xi_n, \hat{\theta}) \log p(z_{n,1}) + \sum_{t=1}^{T_n-1} \log \mathcal{P}(s_{n,t+1}|s_{n,t}, a_{n,t}) \quad (1)$$

$$+ \sum_{t=1}^{T_n} \sum_{z_{n,t}} p(z_{n,t}|\xi_n, \hat{\theta}) \log \pi_{z_{n,t}}(a_{n,t}|s_{n,t}^L; r_z) \quad (2)$$

$$+ \sum_{t=1}^{T_n-1} \sum_{z_{n,t}, z_{n,t+1}} p(z_{n,t}, z_{n,t+1}|\xi_n, \hat{\theta}) \log \mathcal{P}_z(z_{n,t+1}|z_{n,t}, s_{n,t}). \quad (3)$$

Here are some remarks: (I) We incorporate state dependency into the hidden mode transition $\mathcal{P}_z$, such that $z_{t+1}$ depends not only on the previous hidden mode $z_t$ but also on the current state $s_t$. This

modification results in longer segments of hidden modes with reduced fast-switching phenomena. (II) If we have estimated the current policy $\pi_{z_n}$ based on the current reward estimate $r_z$, we can apply any inference method to estimate the posterior probabilities $p(z_{n,t}|\xi_n, \hat{\theta})$ and $p(z_{n,t}, z_{n,t+1}|\xi_n, \hat{\theta})$. In this work, we use the standard forward-backward message-passing algorithm. (III) It is important to note that $\mathcal{P}(s_{n,t+1}|s_{n,t}, a_{n,t})$ represents the environment transition and is not involved in the optimization process. The detailed derivation can be found in Appendix A.1.

---

**Algorithm 1** The SWIRL Algorithm

---

**Data:** Expert demonstrations $\mathcal{D} = \{\xi_1, \xi_2, \ldots, \xi_N\}$
**Result:** The posterior probabilities of hidden modes $z$, rewards $r_z$ for each mode, and other parameters in $\theta$
Initialize parameters $\theta^0$
**for** $k = 1, 2, \ldots, K$ **do**
    **E-step**
        For each hidden mode $z$ and corresponding reward $r_z^k$, compute the soft $Q$-function using Eq. 4 for $I$ iterations. Compute the policy with the Boltzmann distribution:

$$\pi_z(a|s) = \frac{\exp\{Q_z^I(s,a)/\alpha\}}{\sum_{a' \in \mathcal{A}} \exp\{Q_z^I(s,a')/\alpha\}}$$

        to obtain $\pi_z^k(a|s^L; r_z^k)$, $\forall s \in \mathcal{S}$.
        For each trajectory $\xi_n$, use forward-backward message passing to calculate $p(z_{n,t}|\xi_n, \theta^k)$ and $p(z_{n,t}, z_{n,t+1}|\xi_n, \theta^k)$.
        Use the posteriors, $\pi_z^k(a|s^L; r_z^k)$, and $\theta^k$ to compute the auxiliary function $G$ (Eqs. 1-3).
    **end**
    **M-step**
        Update parameters $\theta$ using gradient descent on the auxiliary function $G$ with learning rate $\eta_k$:

$$\theta^{k+1} \leftarrow \theta^k - \eta_k \nabla_\theta G(\theta, \theta^k).$$

    **end**
**end**

---

Consequently, to fully compute the auxiliary function, we must calculate $\pi_z(a|s^L; r_z)$ in Eq. 2, which represents the current optimal policy based on the reward estimate $r_z$ for every hidden mode $z$. This term represents the objective function for optimizing the reward estimate $r_z$ during the M-step. To parameterize the policy in terms of the reward, we use Soft-Q iteration (Haarnoja et al., 2017). Specifically, for the $i$-th iteration, the Q function will be updated through

$$Q^{i+1}(s,a) \leftarrow r_z(s,a) + \alpha\gamma \log \sum_{a' \in \mathcal{A}} \exp\{Q^i(s',a')/\alpha\}, \tag{4}$$

where $\alpha$ is a predefined temperature parameter. The policy $\pi_{z_{n,t}}(a_{n,t}|s_{n,t}^L; r_z)$ in Eq. 2 is derived from a Boltzmann distribution of the computed $Q$ function, making it a differentiable function of the reward function $r_z$. In the M-step, to maximize the auxiliary function $G$, we compute the gradient of $G$ with respect to $r_z$ through the differentiable policy term, and with respect to all other parameters in $\theta$ in other objective terms. The inference procedure alternates between the E-step and M-step until convergence or a predetermined number of iterations. The algorithm is summarized in Algorithm 1.

### 3.5 CONNECTION TO DYNAMICS-BASED BEHAVIOR ANALYSIS METHODS

Traditional methods for analyzing animal behavior in neuroscience often use autoregressive dynamics models, with the autoregressive hidden Markov model (ARHMM) being the most prevalent (Wiltschko et al., 2015; Weinreb et al., 2024). ARHMMs assume that the animal behavior consists of multiple segments represented by a hidden Markov model, where each segment evolves through an autoregressive process. Using the notation established earlier, we denote hidden modes as $z_t$ at time $t$, following the transition $p(z_{t+1}|z_t)$. At each time step $t$, the observation state $s_t$ follows conditionally linear (or affine) dynamics, determined by the discrete mode $z_t$. This can be expressed as $s_{t+1} = A_{z_t} s_t + v_t$, where $A_{z_t}$ is the linear dynamics associated with $z_t$ and $v_t$ represents Gaussian noise. If $z_t$ changes, the linear dynamics will also change accordingly. More generally, we

can represent the dynamics as $p(s_{t+1}|s_t, z_t)$. Consequently, the overall generative model for the ARHMM can be summarized as follows: (1) $z_t \sim p(z_t|z_{t-1})$, and (2) $s_{t+1} \sim p(s_{t+1}|s_t, z_t)$. Let's outline the generative model of SWIRL without history dependency: (1) $z_t \sim p(z_t|z_{t-1})$, and (2) $s_{t+1} \sim \sum_{a_t} p(s_{t+1}|s_t, a_t)\pi(a_t|s_t, z_t)$. The term $\pi(a_t|s_t, z_t)$ arises because the policy is derived from $r_{z_t}$. Consequently, the primary distinction between ARHMM and SWIRL lies in the dynamics used to generate $s_{t+1}$.

We can show that SWIRL is a more generalized version of ARHMM. In a deterministic MDP, where $p(s_{t+1}|s_t, a_t)$ is a delta function and each action $a_t$ uniquely determines $s_{t+1}$, $s_{t+1}$ directly implies $a_t$. Thus, $\sum_{a_t} p(s_{t+1}|s_t, a_t)\pi(a_t|s_t, z_t) = \pi(a_t|s_t, z_t) = p(s_{t+1}|s_t, z_t)$. This effectively reduces SWIRL to ARHMM. In the second real-world experiment, the MDP setup satisfies these assumptions. In such a case, ARHMM can be seen as performing policy learning through behavioral cloning without learning a reward function, whereas SWIRL employs IRL to learn the policy.

Having established this connection, we can view SWIRL as a more generalized version of ARHMM, as it permits the MDP to be stochastic and allows multiple actions to result in the same preceding state. Additionally, explicitly modeling the policy introduces a reinforcement learning framework that better represents the decision-making processes of animals and reveals the underlying reward function. For SWIRL with history dependency, we can further connect it to the recurrent ARHMM (Linderman et al., 2016), which expands $p(z_{t+1}|z_t)$ to $p(z_{t+1}|z_t, s_t)$.

An advanced version of the ARHMM is the switching linear dynamical system (SLDS), which assumes that the state $s_t$ is unobserved. Instead, the observed variable $y_t$ is a linear transformation of $s_t$. Thus, the complete generative model for SLDS consists of: (1) $z_t \sim p(z_t|z_{t-1})$, (2) $s_{t+1} \sim p(s_{t+1}|s_t, z_t)$, and (3) $y_{t+1} \sim p(y_{t+1}|s_{t+1})$. This suggests that the representation $s_t$ capturing the primary dynamics is, in fact, a latent representation of the external world $y_t$. Building on this concept, we can extend SWIRL into a latent variable model, where $s_t$ serves as the latent representation of the true observation state $y_t$. This corresponds to the setup of Partial Observation Markov Decision Processes (POMDPs) in the literature. This extension will link SWIRL to representation learning in reinforcement learning, which we plan to explore further in future work.

Thus, we argue that SWIRL offers a more generalized and principled approach to studying animal behavior compared to commonly used dynamics-based models, as one can draw inspiration from the development of (latent) dynamics models to enhance advanced IRL methods for analyzing animal decision-making processes.

## 4 RESULTS

Throughout the experiment section, we use the following terminology to denote our proposed algorithms and the baseline models we compare.
• **MaxEnt** (Ziebart et al., 2008; 2010): Maximum Entropy IRL where the reward function is only a function of the current state and action. It is a single-mode IRL approach with a single reward function.
• **Multi-intention IQL** (Zhu et al., 2024): learns time-varying reward functions based on HM-MDP. It is a SWIRL model with no history dependency.
• **Locally Consistent IRL** (Nguyen et al., 2015): learns time-varying reward functions based on HM-MDP. It is a SWIRL model with no action-level history dependency.
• **ARHMM** (Wiltschko et al., 2015): learns the segmentation of animal behaviors using autoregressive dynamics combined with a hidden Markov model.
• **rARHMM** (Linderman et al., 2016): recurrent ARHMM whose transition probability of the hidden modes also relies on the state.
• **I-1, I-2**: the baseline variant of our proposed SWIRL method which assumes the transition kernel $\mathcal{P}_z$ is **independent** of the state. The reward and policy depend either on the current state (in the case of I-1) or on both the current and previous states (in the case of I-2). Note that I-1 represents the simplest version of SWIRL, which corresponds to Multi-intention IQL. Thus, we use I-1 to denote Multi-intention IQL. The model can incorporate an arbitrary history length $L$ for the policy and reward; in this paper, we use $L = 1$ and $L = 2$.
• **S-1, S-2**: our proposed SWIRL method where $\mathcal{P}_z$ is **state dependent**. The suffix follows the same setup as above. S-1 corresponds to Locally Consistent IRL.

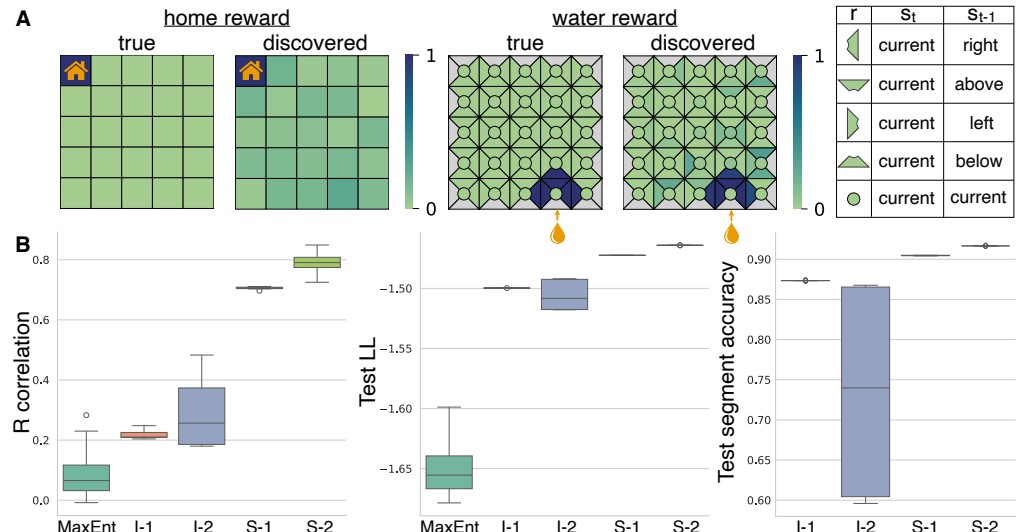

Figure 2: **Simulation experiment on a 5 × 5 gridworld.** (A) Comparison between the true and discovered reward maps. The color scale represents reward values ranging from 0 to 1. The home reward is defined as $r(s_t)$, while the water reward depends on both the current and previous locations, $r(s_t, s_{t-1})$. To present the water reward, each location is divided into five groups, as detailed in the table on the far right. For example, the polygon in the first row represents the reward value when $s_t$ is the current location and $s_{t-1}$ is the location to the right. Light grey indicates an impossible transition where no reward exists. (B) Box plots illustrating the Pearson correlation between the true and recovered reward maps, test log-likelihood, and test segmentation accuracy. The x-axis represents the five different models. Outlier selection method is described in Appendix B.6.

### 4.1 APPLICATION TO A SIMULATED GRIDWORLD ENVIRONMENT

We begin by testing our method on simulated trajectories within a 5 × 5 gridworld environment, where each state allows for five possible actions: up, down, left, right, and stay. The agent alternates between two reward maps: a home reward map and a water map (see Fig. 2A). Following the design of real animal experiments (Rosenberg et al., 2021), we assume that the water port provides water to the agent only once per visit. Therefore, under the water reward map, the agent receives a reward for (1) visiting the water state if it was not in the water state previously or (2) leaving the water state. The home reward map returns a reward at the home state. This leads to a non-Markovian reward function that relies on both the current state and the previous state. We employed soft-Q iteration to determine the optimal policy for each reward function and generated 200 trajectories based on the learned policy, using a history-dependent hidden-mode switching dynamic $\mathcal{P}_z(z_{t+1}|z_t, s_t)$. Accordingly, the agent is more likely to switch to the home map after visiting the water port and to switch to the water map after returning home. Each trajectory consists of 500 steps.

We then used SWIRL to learn the reward functions and the transition dynamics between them, based on 80% of the generated trajectories. As a baseline, we employed the Maximum Entropy IRL (MaxEnt) method and tested four variations of the SWIRL models (I-1, I-2, S-1, S-2), with I-1 representing multi-intention IQL. Fig. 2A displays a comparison between the true and discovered reward functions, while Fig. 2B presents boxplots showing the Pearson correlation between the true and recovered reward functions, along with the test log-likelihood (LL) and test segmentation accuracy (which measures the ability to predict the correct segments for home and water modes). The test performance was evaluated using the remaining 20% of the trajectories. Notably, accurate reward recovery was only achieved with the S-2 model. All four SWIRL variations outperformed MaxEnt, indicating the presence of more than one hidden model. Both the state dependency of hidden-mode transitions (decision-level dependency) and the history dependency reward function (action-level dependency) contributed to further improvements in test LL and segmentation accuracy. Specifically, only the state-dependent models (S-1, S-2) could accurately and robustly recover test segments, while the independent models (I-1, I-2) exhibited lower accuracy with higher variance. This is attributed to the non-Markovian reward design, where the agent can only receive water once per visit. Notably, S-2, the full SWIRL model incorporating both decision-level and action-level dependencies, demonstrated the best performance across all metrics.

### 4.2 APPLICATION OF SWIRL TO LONG, NON-STEREOTYPED MOUSE TRAJECTORIES

We then applied SWIRL to the long, non-stereotyped trajectories of mice navigating a 127-node labyrinth environment with water restrictions (Rosenberg et al., 2021). In this experiment, a cohort of 10 water-deprived mice moved freely in the dark for 7 hours. A water reward was provided at an end node (Fig. 3A), but only once every 90 seconds at most. Similar to the simulated experiment, the 90-second condition forces the mice to leave the port after drinking water, leading to a non-Markovian internal reward function. For our analysis, we segmented the raw node visit data into 238 trajectories, each comprising 500 time points. This data format presents a considerably greater challenge compared to the same dataset processed with more handcrafted methods in previous IRL applications (Ashwood et al., 2022a; Zhu et al., 2024), which were limited to clustered, stereotyped trajectories of only 20 time points in length.

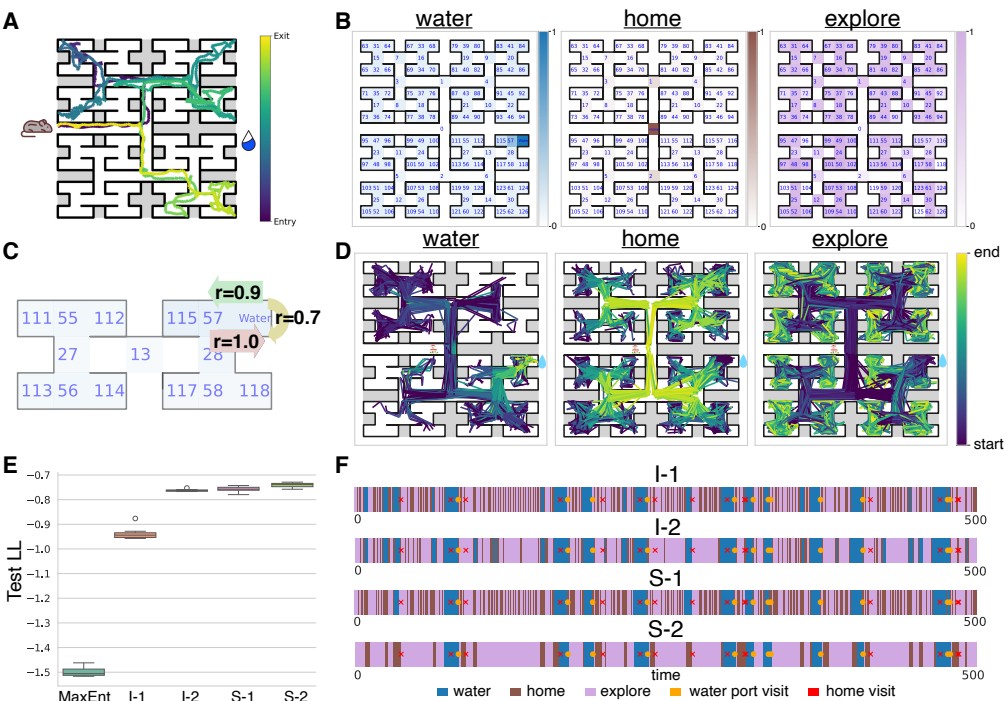

Figure 3: **Water-restricted labyrinth experiment.** (A) Setup for the labyrinth experiment. (B) Inferred reward maps from SWIRL (S-2) under three hidden modes: water, home, and explore. To enhance visualization, the inferred reward $r(s_t, s_{t-1})$ was averaged over $s_{t-1}$ to produce $r(s_t)$. (C) History dependency inferred by SWIRL (S-2), as reflected in the reward map for the water mode. (D) Trajectories segmented into hidden modes based on SWIRL (S-2) predictions. (E) Boxplot showing held-out test LL, with the x-axis representing the five different models. Outlier selection method is described in Appendix B.6. (F) Segments of a trajectory from held-out test data, predicted by four SWIRL models. The orange dot indicates when the mouse visits the water port, while the red cross denotes the mouse's visit to state 0 (home) at that time.

### 4.2.1 SWIRL INFERRED INTERPRETABLE HISTORY-DEPENDENT REWARD MAPS

We applied SWIRL to 80% of the 238 mouse trajectories from the water-restricted labyrinth experiments. According to Rosenberg et al. (2021), mice quickly learned the labyrinth environment and began executing optimal paths from the entrance to the water port within the first hour of the experiment. Therefore, we assume the mice acted optimally concerning the internal reward function guiding their behavior. Fig. 3E displays the held-out test LL for MaxEnt and the SWIRL variations based on the remaining 20% of trajectories. The state dependency in hidden-mode switching dynamics and the history dependency in the reward function contributed to improved test performance. The final SWIRL model (S-2) successfully inferred a water reward map, a home reward map, and an explore reward map (Fig. 3B). For better visualization, we averaged the S-2-recovered history-dependent rewards across previous states and normalized the reward values to a range of (0, 1). In the water reward map, mice received a high reward for visiting the water port. In the home

reward map, there was a high reward for visiting state 0 at the center of the labyrinth, which also served as the entrance and exit. Mice occasionally went to state 0 to enter or leave the labyrinth and sometimes passed by on their way to other nodes. In the explore reward map, mice received a high reward for exploring areas of the labyrinth other than state 0 and the water port.

We are particularly excited to have inferred an interpretable history-dependent reward map for the water port (Fig. 3C). It indicates that mice receive a high reward (1.0) for reaching the water port when their previous location was not the water port. If their prior location was the water port, there is still a reward (0.7) for staying there, but the reward for leaving the water port is even higher (0.9). This observation aligns with the water port design, as mice can only obtain water once every 90 seconds. Consequently, it makes sense that the mice would want to leave the water port after reaching it. Such insights would not be captured by a Markovian reward function that depends solely on the current state.

### 4.2.2 SWIRL INFERRED INTERPRETABLE HISTORY-DEPENDENT HIDDEN-MODE SEGMENTS

We then visualized all mouse trajectories based on the hidden-mode segments predicted by SWIRL (S-2) (Fig. 3D). In segments classified as water mode, mice start from various locations in the labyrinth and move toward the water port. In segments identified as home mode, mice begin from distant nodes and head toward the center of the labyrinth (home). In segments categorized as explore mode, mice start from junction nodes or the water port and explore end nodes other than the water port. This result demonstrates that SWIRL can identify sub-trajectories of varying lengths from raw data spanning 500 time points, allowing us to visualize them together and reveal clustered behavioral strategies. This capability has not been achieved by previous studies on freely moving animal behavior over extended recording periods, and we conducted this analysis without prior knowledge of the locations of the water port or home.

We also provide a detailed visualization of the hidden-mode segments from an example trajectory in the held-out test data and compare the segmentation performance of the four SWIRL variations (Fig. 3F). In the S-2 segments, visits to the water port (indicated by orange dots) consistently occur at the end of a water mode segment, while visits to state 0 (home) (indicated by red crosses) typically happen at the conclusion of a home mode segment. Notably, home mode segments that do not include a visit to state 0 can still be valid, as these segments may end at state 1 or 2 (see Appendix C.1). In contrast, the I-1, I-2, and S-1 segments exhibit instances of water segments that do not involve a visit to the water port, along with many home segments that lack clear interpretability. Overall, S-2 successfully identifies robust segments of reasonable length, avoiding the numerous rapid switches seen in the other variations. We attribute this to both the state dependency of hidden mode transitions and the history dependency in rewards. This suggests that mice are unlikely to make quick changes in their decisions; instead, they make choices based on their current location and take into account at least two locations while navigating the maze.

### 4.3 APPLICATION OF SWIRL TO MOUSE SPONTANEOUS BEHAVIOR TRAJECTORIES

We also employed SWIRL on a dataset in which mice wandered an empty arena without explicit rewards (Markowitz et al., 2023). In this experiment, mouse behaviors were recorded via depth camera video, and dopamine fluctuations in the dorsolateral striatum were monitored. The dataset includes behavior "syllables" inferred by MoSeq (Wiltschko et al., 2015), which indicate the type of behavior exhibited by the mice during specific time periods (e.g., grooming, sniffing, etc.). Consequently, the trajectories consist of behavioral syllables, with each time point representing a syllable. We selected 159 trajectories, each comprising 300 time points, by retaining only the 9 most frequent syllables and merging consecutive identical syllables into a single time point. This method, also used in previous reinforcement learning studies on this dataset (Markowitz et al., 2023), ensures that each syllable has sufficient data for learning and allows the model to concentrate on the transitions between different syllables.

The MDP for this experiment comprises 9 states and 9 actions, where the state represents the current syllable and the action signifies the next syllable. As mentioned in Section 3.5, the ARHMM can be viewed as a variant of SWIRL that learns the policy through behavior cloning. In other words, the policy for this MDP aligns with the emission probability of the ARHMM. This setup offers an excellent opportunity to compare the performance of SWIRL with ARHMM and its variant, rARHMM.

We applied SWIRL, rARHMM, ARHMM, and MaxEnt to 80% of the trajectories and assessed the held-out test LL on the remaining 20% (Fig. 4B). All four SWIRL models outperformed ARHMM and rARHMM on this dataset, indicating that learning rewards is more beneficial for behavior segmentation and explaining the behavior trajectories. Interestingly, the history dependency in the reward function resulted in lower test LL, as S-1 and I-1 demonstrated higher test LL than S-2 and I-2. We believe this is attributable to the merging of consecutive identical syllables and the selection of the top 9 syllables during the preprocessing phase for this dataset. As a result of these steps, the actual time interval between $s_{t-1}$ and $s_t$ may vary significantly, leading to a poorly defined time concept that complicates the model's ability to capture the history dependency in the reward function. However, we can use SWIRL with different variations as a hypothesis-testing tool. The variation yielding the highest test LL may be regarded as more accurately reflecting the dynamics and structure of the data. Consequently, these results suggest that the behavior trajectories exhibit only Markovian dependency rather than long-term non-Markovian dependency. Since S-1 remains higher than I-1, we conclude that the state dependency in the hidden mode transition contributes to explaining the data. Furthermore, as discussed in Appendix C.2, SWIRL recovered reward maps and hidden-mode segments provide insights into the variability of dopamine impacts on animal spontaneous behavior.

While the non-Markovian action-level history dependency introduced by SWIRL does not demonstrate superior performance in this particular experiment, the findings showcases SWIRL's unique contribution to neuroscience research. Specifically, SWIRL serves as a powerful tool for hypothesis testing in behavioral datasets, enabling researchers to validate or challenge hypotheses regarding decision-level dependency as well as non-Markovian action-level dependency. This versatility further confirms SWIRL's great potential in advancing our understanding of complex behaviors.

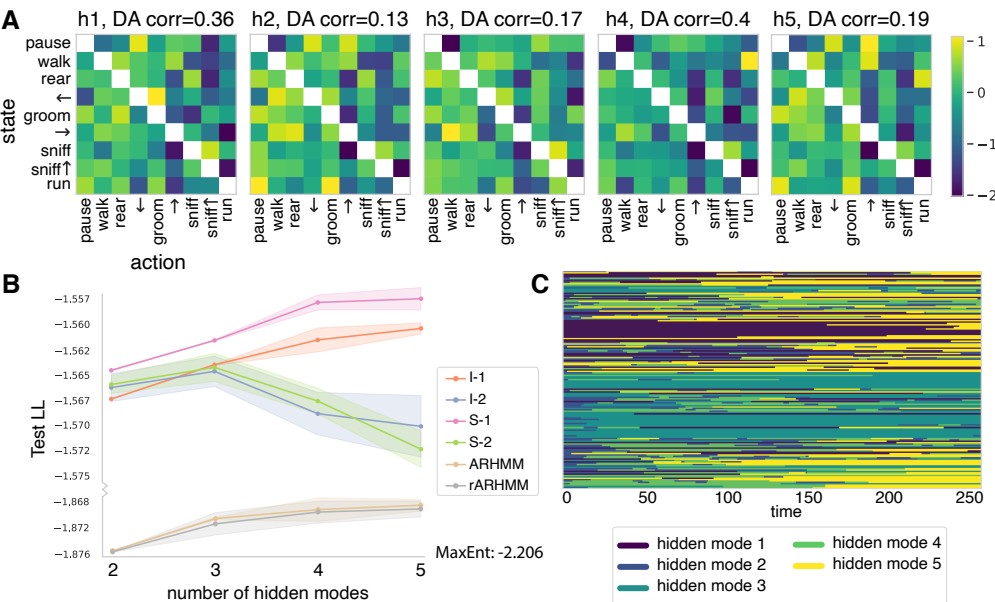

Figure 4: **Mouse spontaneous behavior experiment.** (A) SWIRL (S-1) inferred z-scored reward maps for five hidden modes. h1 denotes hidden mode 1, and so on. DA corr represents the Pearson correlation between the inferred reward map and the averaged dopamine fluctuation levels. (B) Held-out test LL for each model across different number of hidden modes. The shaded area represents the total area that falls between one standard deviation above and below the mean. (C) Inferred hidden-mode segments for all trajectories, with each row representing a trajectory.

## 5 DISCUSSION

We introduce SWIRL, an innovative inverse reinforcement learning framework designed to model history-dependent switching reward functions in complex animal behaviors. Our framework can infer interpretable switching reward functions from lengthy, non-stereotyped behavioral tasks, achieving reasonable hidden-mode segmentation—a feat that, to the best of our knowledge, has not been accomplished previously.

## REPRODUCIBILITY STATEMENT

SWIRL codes can be found at the following anonymous repository: `https://anonymous.4open.science/r/SWIRL-86F6`. Both the labyrinth dataset (Rosenberg et al., 2021) and the spontaneous behavior dataset (Markowitz et al., 2023) are publicly available and can be accessed through the data repositories provided in their respective original publications.

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

# A   APPENDIX A

## A.1   DERIVATION OF SWIRL OBJECTIVES BY EM ALGORITHM

Here we need to learn $\theta \triangleq (r_z, \mathcal{P}_z, p(s_1), p(z_1))$. Note that the total probability for a sequence $\{(s_t, a_t, z_t)\}_{t=1:T}$ is

$$\log p(z, s, a) = \log p(z_1) p(s_1) \pi_{z_1}(a_1|s_1; r_z) \prod_{t=2}^{T} \mathcal{P}(s_t|s_{t-1}, a_{t-1}) \mathcal{P}_z(z_t|z_{t-1}, s_{t-1}) \pi_{z_t}(a_t|s_t^L; r_z).$$

The expectation across all possible sequences is given by

$$\mathbb{E}[\log p(z, s, a)] = \sum_{n=1}^{N} \log p(s_{n,1}) + \sum_{n=1}^{N} p(z_{n,1}|s_1, a_1) \log p(z_{n,1})$$

$$+ \sum_{n=1}^{N} \sum_{t=2}^{T} \log \mathcal{P}(s_{n,t}|s_{n,t-1}, a_{n,t-1})$$

$$+ \sum_{n=1}^{N} \sum_{t=2}^{T} p(z_{n,t}, z_{n,t-1}|\xi_n) \log \mathcal{P}_z(z_{n,t}|z_{n,t-1}, s_{n,t-1})$$

$$+ \sum_{n=1}^{N} \sum_{t=1}^{T} p(z_{n,t}|\xi_n) \log \pi_{z_{n,t}}(a_{n,t}|s_{n,t}^L; r_z).$$

Now we use Expectation-Maximization to find $\theta$. E step:

$$G(\theta, \hat{\theta}) = \sum_z p(z|\xi_{1:N}, \hat{\theta}) \log p(z, \xi_{1:N}, |\hat{\theta})$$

$$= \sum_z \left( \prod_{n=1}^{N} p(z_{n,1:T}|\xi_n, \hat{\theta}_n) \right)$$

$$\sum_{n=1}^{N} \left\{ \log p(s_{n,1}) + \log p(z_{n,1}) + \sum_{t=1}^{T_n} (\log \pi_{z_{n,t}}(a_{n,t}|s_{n,t}^L; r_z)) + \sum_{t=1}^{T_{n-1}} (\log \mathcal{P}(s_{n,t+1}|s_{n,t}, a_{n,t})$$

$$+ \log \mathcal{P}_z(z_{n,t+1}|z_{n,t}, s_{n,t})) \right\}$$

$$= \sum_{n=1}^{N} \log p(s_{n,1})$$

$$+ \sum_{n=1}^{N} \sum_z p(z_{n,1} = z|\xi_n, \hat{\theta}) \log p(z_{n,1})$$

$$+ \sum_{n=1}^{N} \sum_{t=1}^{T_n} \sum_z p(z_{n,t} = z|\xi_n, \hat{\theta}) \log \pi_{z_{n,t}}(a_{n,t}|s_{n,t}^L; r_z)$$

$$+ \sum_{n=1}^{N} \sum_{t=1}^{T_n-1} \sum_z \sum_{z'} p(z_{n,t} = z, z_{n,t+1} = z'|\xi_n, \hat{\theta}) \log \mathcal{P}_z(z_{n,t+1}|z_{n,t}, s_{n,t})$$

$$+ \sum_{n=1}^{N} \sum_{t=1}^{T_n-1} \log \mathcal{P}(s_{n,t+1}|s_{n,t}, a_{n,t}).$$

M step:

$$\theta^{k+1} = \arg\max_{\theta} G(\theta, \theta^k).$$

For notational simplicity, we only consider a specific trajectory $n$. To compute $G(\theta, \theta^k)$, we need to estimate

$$p(z_t|s_{1:T}, a_{1:T}, \hat{\theta})$$

and

$$p(z_t = z, z_{t+1} = z'|s_{1:T}, a_{1:T}, \hat{\theta}).$$

Thus we can use message passing algorithm, where we define the forward-backward variables $\alpha$ and $\beta$. Forward variables $\alpha_{t,z}$, for $t = 1, \ldots, T$:

$$\alpha_{1,z} = p(z_1 = z|\hat{\theta}),$$

$$\alpha_{t,z} = p(s_{1:t}, a_{1:t}, z_t = z|\hat{\theta})$$

$$= \sum_{z'} p(s_{1:t-1}, a_{1:t-1}, z_{t-1} = z'|\hat{\theta})\mathcal{P}_z(z_t = z|s_{t-1}, z_{t-1} = z')p(s_t|s_{t-1}, a_{t-1})\pi_{z_t}(a_t|s_t^L; r_z)$$

$$= \sum_{z'} \alpha_{t-1,z'}\mathcal{P}_z(z_t = z|s_{t-1}, z_{t-1} = z')p(s_t|s_{t-1}, a_{t-1})\pi_{z_t}(a_t|s_t^L; r_z).$$

Backward variables $\beta_{t,z}$, for $t = 1, \ldots, T$:

$$\beta_{T,z} = 1,$$

$$\beta_{t,z} = p(s_{t+1:T}, a_{t+1:T}|s_t, a_t, z_t = z, \hat{\theta})$$

$$= \sum_{z'} \beta_{t+1,z'}p(s_{t+1}|s_t, a_t)\pi_{z'}(a_t|s_t^L; r_z)\mathcal{P}_z(z_{t+1} = z'|z_t = z, s_{t+1}),$$

$$\beta_{1,z} = p(s_{1:T}, a_{1:T}|z_1 = z, \hat{\theta})$$

$$= \sum_{z'} \beta_{1,z'}p(s_1)\pi_{z'}(a_1|s_1^L; r_z)\mathcal{P}_z(z_1 = z'|z_0 = z, s_1).$$

Therefore,

$$p(z_t = z|\xi, \hat{\theta})$$

$$= p(z_t = z, \xi|\hat{\theta})/p(\xi|\hat{\theta})$$

$$= p(s_{1:t}, a_{1:t}, z_t = z|\hat{\theta})p(s_{t+1:T}, a_{t+1:T}|s_t, a_t, z_t = z, \hat{\theta})/p(\xi|\hat{\theta})$$

$$= \alpha_{t,z}\beta_{t,z}/p(\xi|\hat{\theta}).$$

Furthermore,

$$p(z_{t-1}, s_{t-1}, z_t|\xi, \hat{\theta}) = p(z_{t-1}, s_{t-1}, z_t, \xi|\hat{\theta})/p(\xi|\hat{\theta})$$

$$= p(s_{1:t-1}, a_{1:t-1}, z_{t-1})p(z_t|z_{t-1}, s_{t-1})p(s_t, a_t|s_{t-1}, a_{t-1}, z_t)p(s_{t+1:T}, a_{t+1:T}|z_t, s_t, a_t)/p(\xi|\hat{\theta})$$

$$= \frac{p(s_{1:t-1}, a_{1:t-1}, z_{t-1})p(z_t|z_{t-1}, s_{t-1})p(s_t|s_{t-1}, a_{t-1})p(a_t|s_t, z_t)p(s_{t+1:T}, a_{t+1:T}|z_t, s_t, a_t)}{p(\xi|\hat{\theta})}$$

$$= \frac{\alpha_{t-1,z_{t-1}}\mathcal{P}_z(z_t|z_{t-1}, s_{t-1})\mathcal{P}_z(s_t|s_{t-1}, a_{t-1})\pi_{z_t}(a_t|s_t^L; r_z)\beta_{t,z_t}}{p(\xi|\hat{\theta})}.$$

And finally,

$$p(\xi|\hat{\theta}) = \sum_z \alpha_{T,z} = \sum_z \alpha_{1,z}\beta_{1,z}.$$

### A.2 DISCUSSION ON THE CONVERGENCE

The SWIRL inference procedure follows the Expectation-Maximization (EM) algorithm, which has a convergence guarantee (Wu, 1983). For inferring the reward function under each hidden mode, SWIRL adopts the Maximum Entropy Inverse Reinforcement Learning (MaxEnt IRL) framework, with Soft-Q iteration serving as the RL inner loop. Both Soft-Q iteration (Haarnoja et al., 2017) and MaxEnt IRL (Zeng et al., 2022) have also been rigorously analyzed for convergence. Therefore, the overall convergence of the SWIRL inference procedure can be established based on above works.

### A.3 COMPLEXITY ANALYSIS

Below, we provide a detailed complexity analysis of SWIRL inference procedure under tabular representation of $r_z(s^L, a)$ and $\mathcal{P}_z(z_{t+1}|z_t, s_t)$.

#### A.3.1 NOTATION

- $N$: Number of expert trajectories.
- $T$: Length of each trajectory.
- $Z = |\mathcal{Z}|$: Number of hidden modes.
- $S = |\mathcal{S}|$: Number of states.
- $A = |\mathcal{A}|$: Number of actions.
- $L$: Length for action-level history dependency.
- $I$: Number of iterations in Soft-Q iteration.
- $P_r$: Number of parameters in the reward function $r$. $P_r = Z \cdot S^L \cdot A$ when $r$ is represented in a tabular form.
- $P_{\mathcal{P}_z}$: Number of parameters in the hidden mode transition probabilities $\mathcal{P}_z(z_{t+1}|z_t, s_t)$. $P_{\mathcal{P}_z} = Z \cdot S \cdot Z$ when $\mathcal{P}_z(z_{t+1}|z_t, s_t)$ is represented in a tabular form.
- $P_\theta = P_r + P_z$: Total number of parameters. In this analysis we omit the initial probability $p(z_1)$ and $p(s_1)$ for simplicity.

#### A.3.2 E-STEP COMPLEXITY

The E-step consists of two main tasks:

1. Computing the policy $\pi_z(a|s^L; z)$ for each hidden mode $z$ by Soft-Q iteration.
   - In each iteration, computing $Q_z^{i+1}(s^L, a)$ requires summing over all actions $a'$, resulting in $O(A^2)$ per $s^L$.
   - Time complexity:
     $$O(Z \cdot I \cdot S^L \cdot A^2)$$
     (Soft-Q iteration over $S^L$ states and $I$ iterations for $Z$ hidden modes).
   - Space complexity:
     $$O(Z \cdot S^L \cdot A)$$
     (only need to store the Q-value for current iteration).

2. Using the forward-backward algorithm to compute posterior probabilities $p(z_t|\xi, \theta^k)$ and $p(z_t, z_{t+1}|\xi, \theta^k)$.
   - Forward and backward computations involve summations over $Z^2$ hidden mode pairs at each time step.
   - Time complexity:
     $$O(N \cdot T \cdot Z^2)$$
     (over all timepoints in all trajectories).
   - Space complexity:
     $$O(N \cdot T \cdot Z)$$
     (need to store $\alpha_{t,z}$ and $\beta_{t,z}$ for each time step $t$, hidden mode $z$, and trajectory).

The total E-step time complexity:
$$O(Z \cdot I \cdot S^L \cdot A^2 + N \cdot T \cdot Z^2).$$

The total E-step space complexity:
$$O(Z \cdot S^L \cdot A + N \cdot T \cdot Z).$$

### A.3.3 M-STEP COMPLEXITY

The M-step updates $\theta = \{r, \mathcal{P}_z\}$ by maximizing the auxiliary function $G(\theta, \hat{\theta})$.

1. Computing the loss for reward function $r$ involves computing the policy by Soft-Q iteration, which has time complexity $O(Z \cdot I \cdot S^L \cdot A^2)$. Since we also need to iterate over all timepoints across all trajectories for all hidden modes in the policy, the total time complexity is:

$$O(Z \cdot I \cdot S^L \cdot A^2 + N \cdot T \cdot Z).$$

2. Computing the loss for hidden mode transition $\mathcal{P}_z$ involves iterating across all timepoints in all trajectories for all hidden modes pairs $(z, z')$. Therefore, the time complexity is:

$$O(N \cdot T \cdot Z^2).$$

The total M-step time complexity:

$$O(Z \cdot I \cdot S^L \cdot A^2 + N \cdot T \cdot Z^2).$$

The total M-step space complexity:

$$O(P_\theta) = O(Z \cdot S^L \cdot A + Z \cdot S \cdot Z).$$

### A.3.4 TOTAL COMPLEXITY PER EM ITERATION

The total time complexity:

$$O\left(Z \cdot I \cdot S^L \cdot A^2 + N \cdot T \cdot Z^2\right).$$

The total space complexity:

$$O\left(Z \cdot S^L \cdot A + N \cdot T \cdot Z + Z^2 \cdot S\right).$$

### A.4 SCALABILITY AND BROADER IMPACT

While the current implementation of SWIRL performs efficiently for typical animal behavior datasets in neuroscience, we acknowledge the need for a more general and scalable implementation to address broader applications.

In its current form, every step of the SWIRL inference procedure, except for the Soft-Q iteration, is compatible with large or continuous state-action spaces. However, the Soft-Q iteration is limited to discrete state-action spaces and can be slow with large state-action space as it has time complexity $O(Z \cdot I \cdot S^L \cdot A^2)$. For moderate discrete state-action cases, we still recommend the Soft-Q iteration, as it provides a robust and accurate approach for the RL inner loop of MaxEnt IRL. Nevertheless, for applications requiring scalability and compatibility with general state-action spaces, alternative methods can be adapted to replace the Soft Q iteration in the RL inner loop. For instance, Soft Actor-Critic (Haarnoja et al., 2018).

A promising future direction is to reformulate the standard MaxEnt IRL $r$-$\pi$ bi-level optimization problem in SWIRL as a single-level inverse Q-learning problem, based on the IRL approach known as IQ-Learn (Garg et al., 2021). This method has has been successfully adapted to large language models training, demonstrating great scalability(Wulfmeier et al., 2024). Additionally, the MaxEnt IRL framework can be viewed in an adversarial learning perspective (Fu et al., 2018). Prior work has explored adversarial IRL within the EM framework for continuous state-action spaces, although it relies on a future-option dependency at the decision level, which is not biologically plausible, and does not account for action-level history dependency (Chen et al., 2023).

These advancements suggest that the SWIRL framework has the potential to handle MDPs with larger and general state-action spaces. This scalability positions SWIRL as a valuable tool not only for computational neuroscience but also for broader interest of the machine learning community.

# B APPENDIX B

## B.1 IMPLEMENTATION DETAILS

We implemented SWIRL in JAX. For all three datasets, we split the trajectories into 80% training data and 20% held-out test data. We conducted each experiment with 20 random seeds and selected the top 10 results based on the log-likelihood (LL) of the **training** data. This approach is a common practice when implementing the EM algorithm as EM is sensitive to initial parameters and can get trapped in a local optimum (Weinreb et al., 2024). It ensures that only the most representative outcomes of each model were used for analysis. We then evaluated the performance on the 20% held-out test data.

## B.2 EMPIRICAL RUNTIME

Our SWIRL implementation leverages the advantages of JAX, including just-in-time (JIT) compilation and vectorization, to achieve high computational efficiency. For S-2 experiments across all three datasets (gridworld, labyrinth, and spontaneous behavior), SWIRL converges within 15–30 minutes on a V100 GPU, which takes 50–100 EM iterations. For longer L, a S-4 experiment on labyrinth with 50 EM iterations take 2-3 hours to finish on a L40S GPU. We switch to a L40S GPU for L=4 due to the V100 GPU's insufficient memory capacity.

## B.3 DISCUSSION ON DISCOUNT FACTOR $\gamma$ AND TEMPERATURE $\alpha$

We set the discount factor $\gamma = 0.95$, a standard choice in RL and IRL literature. For the mouse spontaneous behavior dataset, we also tested a smaller $\gamma = 0.7$, as previous literature (Markowitz et al., 2023) suggested this value as optimal for the dataset. However, we observed that the results learned by SWIRL were highly similar for both discount factors, indicating that the choice of $\gamma$ had minimal impact on performance in this case.

We searched for the optimal temperature $\alpha$ in $\{0.01, 0.1, 0.5, 1\}$. For the labyrinth dataset, smaller values of $\alpha$ led to better results for certain hidden modes. This observation aligns with the deterministic nature of behaviors in the labyrinth's tree-like structure. On the contrary, for the spontaneous behavior dataset, where animals exhibit more stochastic behavior patterns, we found higher values of $\alpha$ were more appropriate.

## B.4 DISCUSSION ON THE NUMBER OF HIDDEN MODES $Z$ AND HISTORY LENGTH L

In this section, we discuss the impact and selection of the number of hidden modes $Z = |\mathcal{Z}|$ and action-level history length L.

### B.4.1 Z IN LABYRINTH EXPERIMENT

We evaluated the test LL of SWIRL models on $Z$ from 2 to 5 and found that the best model (S-2) plateaus beyond $Z = 4$ (Fig. 5A). However, $Z = 4$ result does not differ much from the $Z = 3$ result: $Z = 4$ result mainly segments the explore mode of $Z = 3$ into two explore modes with similar reward maps (Fig. 5BC). As a result, we still present $Z = 3$ as the primary result for simplicity.

### B.4.2 L IN LABYRINTH EXPERIMENT

With $Z = 3$, we evaluated the test LL of SWIRL models on $L$ from 1 to 4 and found that the $L = 4$ (S-4) provides the best test LL (Fig. 6A). $L = 3$ and $L = 4$ provide similar hidden segments and reward maps (when averaged over $(s_{t-1}, ...s_{t-L+1})$ to produce $r(s_t)$) as $L = 2$ (Fig. 6BC). In the main paper, we present $L = 2$ (S-2) as the primary result as it has effectively demonstrated the benefits of incorporating non-Markovian action-level history dependency into SWIRL. However, we note that the test LL results in Fig. 6A suggest the presence of longer action-level history dependency ($L > 2$) in this labyrinth dataset. This observation aligns with the partially observable nature of this 127-node labyrinth: The mouse may not know the whole environment, so it tends to rely on longer state history to inform its decision-making. Due to the mouse's limited knowledge of the entire environment, it likely relies on a longer history of prior states to guide its decision-making.

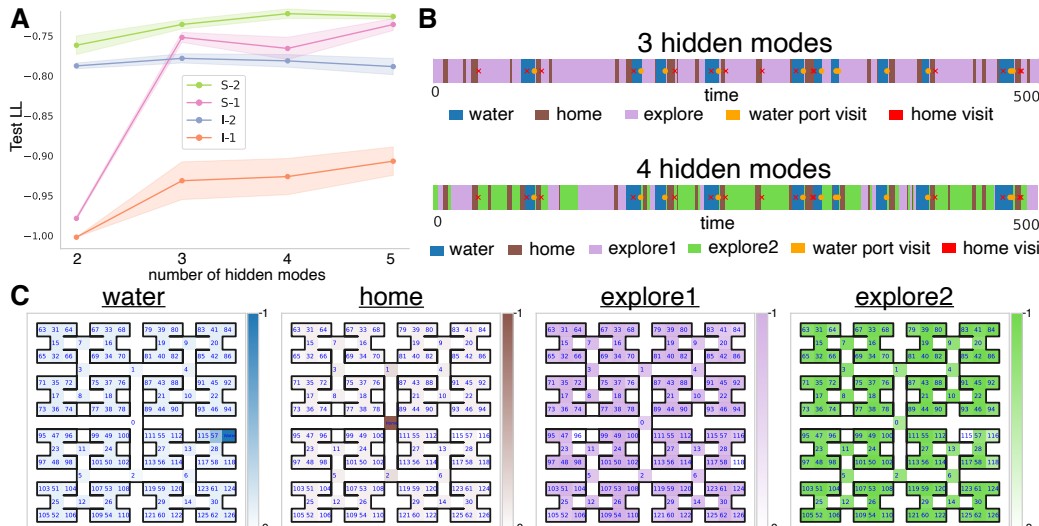

Figure 5: **Water-restricted labyrinth experiment with different number of hidden modes** $Z$.
(A) Held-out test LL for each model across different number of hidden modes. The shaded area represents the total area that falls between one standard deviation above and below the mean. (B) Segments of a trajectory from held-out test data, predicted by SWIRL (S-2) with $Z = 3$ and $Z = 4$. The orange dot indicates when the mouse visits the water port, while the red cross denotes the mouse's visit to state 0 (home) at that time. (C) Inferred reward maps from SWIRL (S-2) with $Z = 4$: water, home, and two explore maps. To enhance visualization, the inferred reward $r(s_t, s_{t-1})$ was averaged over $s_{t-1}$ to produce $r(s_t)$.

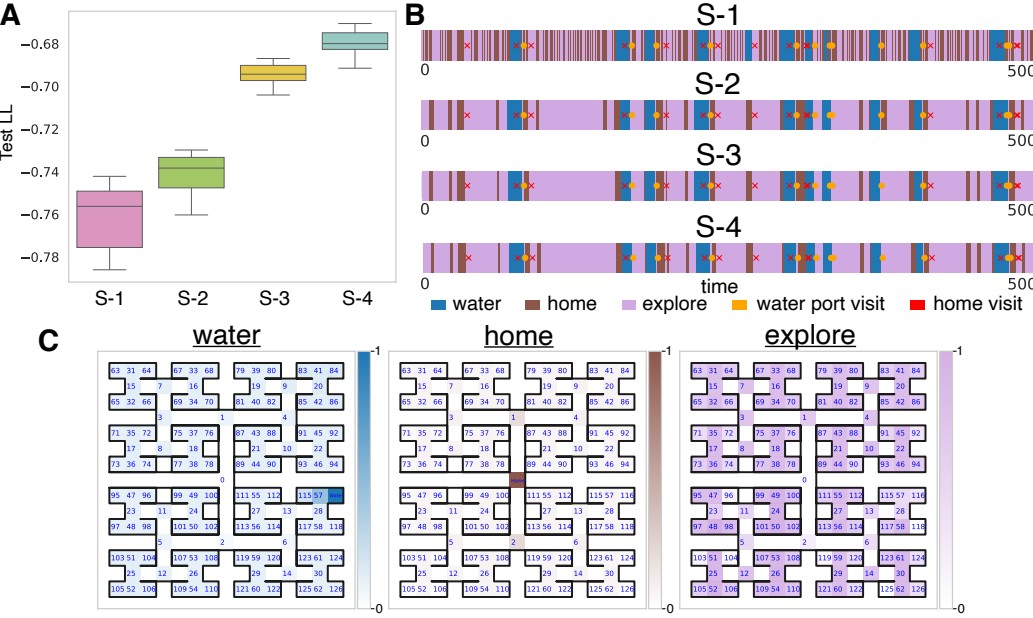

Figure 6: **Water-restricted labyrinth experiment with different action-level history length** $L$.
(A) Boxplot showing held-out test LL, with the x-axis representing the four different models from $L = 1$ to $L = 4$. Outlier selection method is described in Appendix B.6. (B) Segments of a trajectory from held-out test data, predicted by the four SWIRL models. The orange dot indicates when the mouse visits the water port, while the red cross denotes the mouse's visit to state 0 (home) at that time. (C) Inferred reward maps from SWIRL (S-4): water, home, and explore. To enhance visualization, the inferred reward $r(s_t, s_{t-1}, s_{t-2}, s_{t-3})$ was averaged over $(s_{t-1}, s_{t-2}, s_{t-3})$ to produce $r(s_t)$.

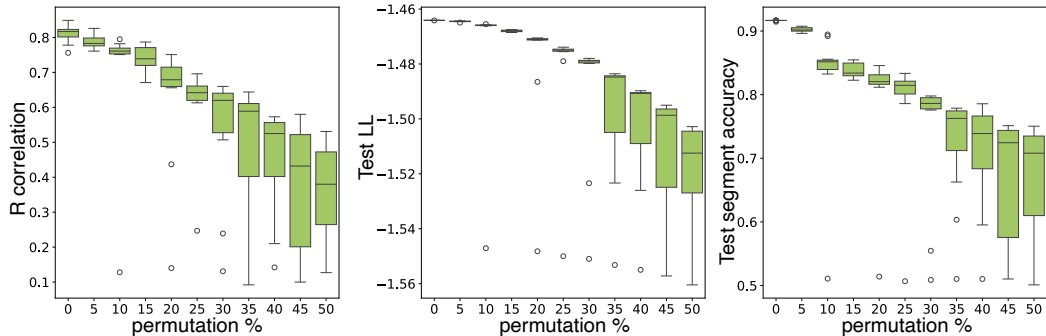

Figure 7: **SWIRL (S-2) experiment on 5 × 5 gridworld dataset with ten random permutations.** Box plots illustrating the Pearson correlation between the true and recovered reward maps, test log-likelihood, and test segmentation accuracy. The x-axis represents the percentage of states and actions permuted in the training data. Outlier selection method is described in Appendix B.6.

### B.4.3   Z AND L IN MOUSE SPONTANEOUS BEHAVIOR EXPERIMENT

In the mouse spontaneous behavior experiment, we find that $L = 1$ (S-1) is the optimal choice, as $L = 1$ (S-1) consistently provides higher test log-likelihood (LL) compared to $L = 2$ (S-2) (Fig. 4B). Additionally, we select $Z = 5$ for the number of hidden modes since the test LL plateaus at $Z = 5$ (Fig. 4B).

### B.5   ROBUSTNESS OF SWIRL

To assess the robustness of SWIRL, we evaluated the performance of SWIRL (S-2) under increasing levels of random perturbations in the simulated gridworld dataset.

Specifically, we introduced random permutations to a percentage of the states and actions in the training data, ranging from 0% to 50%. As expected, performance decreased as the level of permutation increased (Fig. 7). The model maintained high accuracy with less than 10% permutation. Between 10% and 30%, SWIRL demonstrated stable performance, achieving reasonable reward correlations and hidden mode segmentation accuracy despite the noise. Permutation beyond 30% led to very noisy data and it became hard for the model to maintain high performance.

These results suggest that SWIRL can tolerate moderate levels of noise or incomplete data, making it suitable for real-world animal behavior datasets where such challenges are common.

### B.6   OUTLIER SELECTION IN BOX PLOT

All box plots in this paper are drawn by `seaborn.boxplot()` with its default outlier selection method. Specifically, the upper quartile (Q3), lower quartile (Q1), and interquartile range (IQR) are calculated. Values greater than Q3+1.5IQR or less than Q1-1.5IQR are considered as outliers.

## C   APPENDIX C

### C.1   AN EXAMPLE LABYRINTH TRAJECTORY

To further explore the hidden mode segments of the trajectory from held-out test data presented in Fig. 3F, we visualized the segments corresponding to each hidden mode in this trajectory in detail (Fig. 8B). The visualization reveals that "home" segments can remain valid even without a visit to state 0, as these segments often instead terminate at state 1 or state 2, which are next to state 0.

### C.2   DISCUSSION ON REWARD MAPS RECOVERED IN SPONATENOUS BEHAVIOR EXPERIMENT

The best SWIRL model (S-1) recovered reward maps and hidden-mode segments provide insights into the variability of dopamine impacts on animal spontaneous behavior: As illustrated in Fig. 4A,

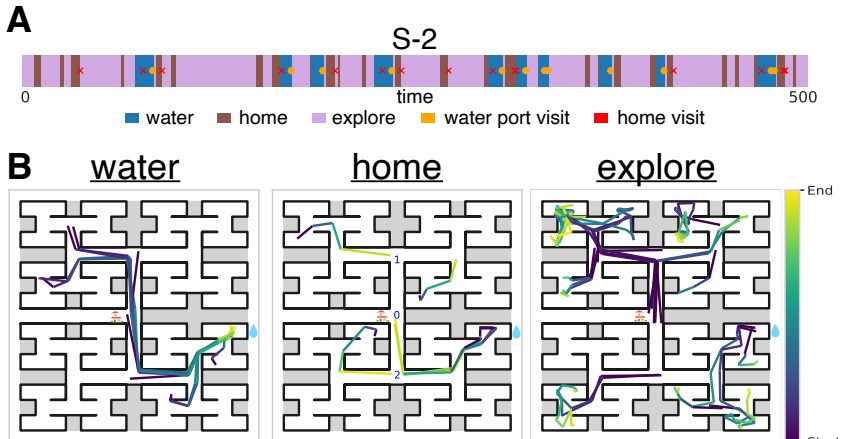

Figure 8: **Hidden mode segments in a labyrinth trajectory.** (A) Segments of a trajectory from held-out test data, predicted by SWIRL (S-2). The orange dot indicates when the mouse visits the water port, while the red cross denotes the mouse's visit to state 0 (home) at that time. (B) The segments of the trajectory shown in (A) are plotted within the labyrinth.

the reward maps exhibit some similarities along with distinct differences. For certain reward maps, there is a decent correlation (e.g., 0.36 and 0.4) with dopamine fluctuations during the corresponding modes. This suggests that dopamine fluctuations can reflect a certain extent of reward during hidden modes 1 and 4. Furthermore, the plot of hidden mode segments across all trajectories reveals identifiable patterns. For instance, hidden mode 2 tends to occur more frequently at the beginning of trajectories, while hidden mode 5 is more prevalent at the end. Previous work by Markowitz et al. (2023) showed that mice are generally more active and move quickly at the start of a trajectory and become slower as they progress. Keeping this in mind, we examined the reward maps in Fig. 4A and found that hidden mode 2 is more rewarding for transitions like run→pause and run→groom, whereas hidden mode 5 offers greater rewards for pause→turn transitions. In comparison, hidden mode 2 is associated with larger movements and more running than hidden mode 5. Similarly, hidden mode 4 encourages transitions from walk to run, which tend to occur more frequently at the beginning of trajectories rather than at the end.

