# OpenReview forum: "Inverse Reinforcement Learning with Switching Rewards and History Dependency for Characterizing Animal Behaviors"
_ICLR.cc/2025/Conference — Submitted to ICLR 2025_

### Official Review · Reviewer_NeTy · 2024-10-28

**Soundness:** 2
**Presentation:** 2
**Contribution:** 1
**Rating:** 3
**Confidence:** 4

**Summary:**

This paper introduces a framework called SWIRL that extends traditional IRL to capture the complex, long-term decision-making processes of animals in natural environments. SWIRL incorporates biologically plausible history dependency at both the decision-level and action-level, allowing it to better account for how past decisions and environmental contexts shape current behavior. The algorithm is evaluated on both synthetic and real-world datasets.

**Strengths:**

This work is well motivated by recent advances in applying IRL methods for characterizing animal complex behavior. By incorporating history-dependent policies and rewards, SWIRL outperformed the state-of-the-art in understanding animal decision-making processes.

**Weaknesses:**

•	Novelty: Insufficient comparison with Nguyen et al.'s 2015 work, which leaves SWIRL's unique contributions unclear.

The difference between SWIRL and the previous work by `Nguyen, Quoc Phong, Bryan Kian Hsiang Low, and Patrick Jaillet. "Inverse reinforcement learning with locally consistent reward functions." Advances in neural information processing systems, 28 (2015)` needs further discussion to clarify the novelty of this work.

Under the similar IRL framework with multiple locally consistent reward functions, Nguyen's algorithm proposed that the transition kernel of reward functions can be dominated by some external inputs. In this case, could SWIRL then be considered as a special case of Nguyen's algorithm with the external input defined as $(s_1, \ldots, s_L)$? Although the RL inner-loop in the two algorithms are slightly different, but I suppose it is not the major difference between SWIRL and Nguyen's work.

It would be helpful if the authors could provide a more detailed comparison between these two algorithms, as well as some additional experiments to compare the performance between them.

•	Unclear scalability of SWIRL, particularly in large environments or with high-dimensional tasks.

•	Lack of transparency in choosing the number of hidden modes across experiments.

•	The math writing of this paper is a bit difficult to follow (see below).

**Questions:**

**Complexity and scalability of the algorithm**

One major issue with variants of MaxEnt IRL methods (even with fixed reward function) is that the computing complexity would be very high since within each loop one needs to solve a forward problem to obtain the policy under the current reward function estimation. This problem would be further scaled up when there are multiple reward functions that needed to be evaluated via solving a RL problem (e.g., in equation (6) of this paper). So my question would be:

1. Is SWIRL guaranteed to converge under some large environments? For example, if the size of the simulated gridworld environment in section 4.1 is not $5 \times 5$ but, say, $50 \times 50$?
2. Following the last question, what is the relationship between computing time of SWIRL and the cardinality of the state, action, and the hidden mode space? This would be helpful to understand what is the maximum capacity of this method, i.e., to which scale of the environment that SWIRL can still handle. To address these questions, it would be helpful if the authors can provide some empirical results on the computing time of SWIRL under different setups.
3. I believe this question would be more suitable for a future work, but just out of curiosity, would SWIRL be possible to integrate some function approximations (of e.g., the state, action, or hidden mode space, since it is now already using gradient method for optimization) so that it can still be applied in high-dimensional tasks?

**Experiments**

1. How did the author choose the number of hidden modes? In the three experiment discussed in this paper, the total number of hidden modes varies across experiment (2 for the first experiment, 3 for the second, and 5 for the last), so I would be curious to know what is the criterion of the authors' to select this hyperparameter. Or did I missed some part such that this number is learnt by the algorithm automatically?
2. The discussion about the reward maps in section 4.3 (lines 492--505) is very hard to follow, as well as the related figures (Figure 4A). In general, I may propose that the results shown in Figures 4A and 4C and corresponding text do not really help for supporting the major claim about SWIRL. This could be due to the lack of space so that lots of experiment details are omitted, and this part would be more suitable for a scientific journal but not a conference. I may recommend the author to change the way of presenting the results.

**Math writing**

Some modifications that would be helpful to increase the readability of the paper are:

1. Numbering of equations in the paper is sloppy, e.g., all equations in the paper are labeled, though some of them are neither referenced nor needed for further discussion. I would recommend labeling only those equations that are referenced later. For example, in lines 660--671, four lines of a single equation is labeled "(8)", but then in lines 685--698, all lines within the same expression are given a different label from "(10)" to "(15)". Or are there any special meanings that I missed?
2. Some display-mode equations are missing punctuation.
3. Line 136, "$R$ corresponds to the reward function $r \in \mathbb{R}$": What is the difference between "$R$" and "$r$" in this notation? According to the rest of the paper (where the notation "$R$" never occurrs again), I would assume they both represent the reward function, then why introducing two different symbol for the same meaning? Besides, the notation "$r \in \mathbb{R}$" defines the reward function "$r$" as a real number, but subsequent reference of "$r$" indicates that it is actually a function on the cartesian product of the state space and action space into the real line. I would recommend make sure the definition of "$r$" is consistent through the paper for a better clarification.

4. Line 170: The symbol "$\mathcal{S}^L$" is used without definition. Does that mean the cartesian product of $L$ state spaces?
5. Line 231, "$\forall s \in \mathcal{S}, z \in Z$": Did you mean "$z \in \mathcal{Z}$"?

6. In the appendix, I only see section A.1.1 but no further sections. Is there are missing section after that? If not, I would suggest remove the subsubsection label for this part.

7. In the derivations in the appendix, the range of some summation is not consistent and not clear. For example, in lines 661--671, the summation is given by "$\sum_{n}$", but in subsequent lines (e.g., line 677) the same summation (I assume) is given by "$\sum_{n = 1}^N$". Is "$\sum_{n}$" just a slang for "$\sum_{n = 1}^N$", or the range in the two sums are actually different?

8. Line 701--703: The notation "$Q(\theta, \theta^k)$" is used without definition. According to the context, I assume it is a typo and the authors would have really meant "$G(\theta, \theta^k)$".


**Minor**

In Figure 3, the color for mode "water" and "explore" is very hard to distinguish if the reader only has access to the printed paper (especially in Figure 3F). Similar issue exists in Figure 4, but not so severe. Consider revising.

--
Post-rebuttal:  I believe the paper can be strongly improved if the concerns about writing, experimental design and representation are addressed, but I will keep my score for now.

---

> ### Author Response · Authors · 2024-11-22
> **Response to Reviewer NeTy**
>
> We sincerely thank the reviewer NeTy for their detailed evaluation and for recognizing our work in advancing the application of IRL methods to model complex animal behaviors. We appreciate their constructive feedback and acknowledgment of SWIRL’s motivation and the improvements it offers in understanding animal decision-making processes through history-dependent policies and rewards.
>
> We are sorry that the contributions of SWIRL may not have been fully recognized by the reviewer and apologize for not making SWIRL's contributions clearer in the paper. Below, we will respond to each concern raised in the Weaknesses and Questions and articulate the contributions of SWIRL:
>
> 1. **Novelty**
>    1.1 **Insufficient comparison with Nguyen et al.'s 2015 work**: We apologize for the oversight. Nguyen et al. 2015 does have a very similar graphical model as SWIRL (S-1) and we have modified our paper to discuss that S-1 results can be used for Nguyen et al. 2015 as a baseline method. Despite this similarity, SWIRL still brings unique contributions to the field of computational neuroscience and the broader machine learning community. A key difference between SWIRL and Nguyen et al. 2015 is the action-level dependency we incorporated. In experiments, we show this action-level dependency is an important perspective in understanding animal behaviors. For instance, in the Labyrinth experiment, the hidden mode segments in Fig. 3.F illustrate that S-1 fails to learn meaningful hidden mode segments, while S-2 succeeds. Additional investigations into S-3 ($L=3$) and S-4 ($L=4$) in Appendix B.4.2 show that higher $L$ values improve test log-likelihoods (LL), suggesting that animals in the labyrinth task rely on longer history for decision-making. In the mouse spontaneous behavior experiment, while longer action-level dependency does not yield better results, the findings reinforce our claim that SWIRL serves as a versatile tool for hypothesis testing in behavioral datasets, enabling researchers to validate or challenge hypotheses regarding decision-level dependency as well as non-Markovian action-level dependency. This adaptability underscores SWIRL’s potential to advance our understanding of complex behaviors.
>
>    1.2 **We would like to further articulate SWIRL contributions here**: As an application-to-neuroscience study submitted to the neuroscience track, SWIRL proposed a framework for IRL with switching reward functions that incorporated biologically feasible decision-level and action-level history dependencies, offering a valuable tool for analyzing animal behavior. SWIRL is the first model to incorporate action-level history dependency, and we have demonstrated its significance in understanding animal behavior, as discussed in the above response 1.1 regarding Nguyen et al.'s 2015 work. Additionally, we are the first to explore the connection between HM-MDPs and other popular dynamics models (e.g., ARHMMs, SLDS) used in animal behavior studies. By bridging these frameworks, we promote a more comprehensive understanding of behavior analysis tools and demonstrate how SWIRL can be adapted to experiments previously analyzed with these models. Moreover, our framework is easily extendable to large and continuous state and action spaces by replacing the Soft-Q iteration in the RL inner loop. This scalability positions SWIRL as not only a valuable tool for computational neuroscience but also a significant contribution to the broader machine learning community. Details on the scalability are described below in response 2.4.

---

> ### Author Response · Authors · 2024-11-22
> **Response to Reviewer NeTy (Cont.)**
>
> 2. **Complexity and scalability of the algorithm**:
>    2.1 **Convergence guarantee**: For a 50x50 gridworld, SWIRL has the convergence guarantee. We have added a convergence analysis discussion in Appendix A.2. The convergence of the SWIRL inference procedure can be established based on the convergence guarantee of the EM algorithm, MaxEnt IRL, and Soft-Q iteration.
>
>    2.2 **Complexity analysis**: We have added a complexity analysis in Appendix A.3. The key bottleneck of SWIRL is in the RL inner loop. Current Soft-Q iteration has a time complexity of $O(ZIS^{L}A^2)$, where $Z$ is the number of hidden modes, $I$ is the number of Soft-Q iterations, $S$ is the number of states, $A$ is the number of actions, and $L$ is the history length. Although the transition model $P(s'|s, a)$ allows us to reduce the $S^L$ space by removing impossible $(s, s')$ pairs, we recognize that longer $L$ in SWIRL does require a considerable amount of computing resources.
>
>    2.3 **Empirical runtime**: We have added an empirical runtime discussion in Appendix B.2. Empirically, SWIRL inference speed is not slow for typical animal behavior datasets. We implemented SWIRL efficiently with JAX, leveraging the features of vectorization and just-in-time compilation to improve our performance. Every SWIRL (S-2) experiment on all three datasets (simulated gridworld, labyrinth, and dopamine) can finish within 15-30 minutes on a V100 GPU. For longer $L$, an S-4 experiment on the labyrinth with 50 EM iterations takes 2-3 hours to finish on an L40S GPU. We switch to an L40S GPU for $L=4$ due to the V100 GPU's insufficient memory capacity.
>
>    2.4 **Integration with function approximators**: Yes, it is possible. We have added a scalability and broader impact discussion in Appendix A.4. While scalability is not the main focus of this work, as typical behavior datasets in neuroscience have small or moderate state-action space and current SWIRL implementation can perform efficiently on them, we acknowledge the need for a more general and scalable implementation to address broader applications. In its current form, every step of the SWIRL inference procedure, except for the Soft-Q iteration, is compatible with large or continuous state-action spaces. However, the Soft-Q iteration is limited to discrete state-action spaces and can be slow with a large state-action space. Nevertheless, for applications requiring scalability and compatibility with general state-action spaces, alternative methods can be adapted to replace the Soft Q iteration in the RL inner loop. For instance, Soft Actor-Critic [1]. Furthermore, a promising future direction is to reformulate the standard MaxEnt IRL $r$-$\pi$ bi-level optimization problem in SWIRL as a single-level inverse Q-learning problem, based on the IRL approach known as IQ-Learn [2]. This method has been successfully adapted to large language models training, demonstrating great scalability [3].
>
> 3. **Lack of transparency in choosing the number of hidden modes across experiments**: We have included discussion on the number of hidden modes with additional experiment results in Appendix B.4. For the labyrinth dataset, in Appendix B.4.1, we tested hidden modes from 2 to 5 and found that the test log-likelihood (LL) plateaus at 4 hidden modes. However, after closely examining the hidden mode segmentation and reward map, we found that the 4 hidden modes result does not differ too much from 3 hidden modes: It mainly segments the explore mode into 2 explore modes with similar reward maps. As a result, we still show 3 hidden modes results in the main paper for simplicity. For the mouse spontaneous behavior experiment, in Appendix B.4.3, we show that we pick 5 hidden modes since the test LL plateaus at 5 hidden modes.
>
> 4. **Spontaneous behavior experiment**:
>    4.1 We apologize for the confusion. The reward discussion was intended to illustrate how SWIRL can provide valuable insights into dopamine studies, as SWIRL is an application-to-neuroscience study. Following the reviewer’s suggestion, we have moved the reward discussion to Appendix C.2 and added a paragraph in the main paper emphasizing how this experiment demonstrates the value of SWIRL’s action-level history dependency, even in light of the lower performance of S-2.
>
>    4.2 While the non-Markovian action-level history dependency introduced by SWIRL (S-2) does not demonstrate superior performance in this particular experiment, the findings showcase SWIRL’s unique contribution to neuroscience research. Specifically, SWIRL serves as a powerful tool for hypothesis testing in behavioral datasets, enabling researchers to validate or challenge hypotheses regarding decision-level dependency as well as non-Markovian action-level dependency. This versatility further confirms SWIRL’s great potential in advancing our understanding of complex behaviors.

---

> > ### Comment · Reviewer_NeTy · 2024-11-25
> >
> > I would like to thank the authors for the detailed rebuttal, most of my concerns are addressed.
> >
> > After reading the other reviewer's comments, I totally agree with the latest comment from reviewer Qvd7 (https://openreview.net/forum?id=v7a4KET0Md&noteId=UtmFHpFX04). Using a different number of random seeds for different experiments does make me concerned.
> >
> > I would agree that a more detailed justification about how the experiments were conducted and a more consistent experimental validation would substantially increase the quality of this paper.
> >
> > Besides, based on the latest revision, I still have the following concerns.
> >
> > **Complexity and scalability**
> >
> > Thank you for the additional section on the complexity analysis of SWIRL. It seems that the cardinality of the state space to the power of the history length $S^L$ is the leading term in the complexity of SWIRL. The authors also mentioned that running the S-4 experiment on the labyrinth with 50 EM iterations takes 2-3 hours, which means that each inner-loop of SWIRL with $S = 127$ and $L = 4$ would need 3 minutes. Then I would expect that applying SWIRL on a $50 \times 50$ gridworld with $L = 4$ would start to be (practically) infeasible, although the authors claimed theoretically that SWIRL is guaranteed to converge at least to some local optimum. (Even assuming that the iteration of EM is still 50, which I think should be increased w.r.t. the environment scale if this is common practice in EM. The authors are welcomed to correct me if this is not the case.)
> >
> > So, my followup question is: would a $50 \times 50$ gridworld still be in the range of the designed application scenario of SWIRL?
> > I believe some empirical results about the run time of SWIRL in different scaled gridworld environments would help the user to understand the maximum scale of the environment that SWIRL can converge within acceptable time.
> >
> > **Minor**
> >
> > 1. It is not standard to label the equations in an algorithm environment (line 233 and 242). Since the involved two equations are not referenced subsequently, I would suggest to unnumber these equations.
> > 2. Line 756, since the authors did not modify the notation "$Q(\theta, \theta^k)$" that I mentioned in the original review, I may expect there is a special meaning that I missed for using a new notation. Could the authors clarify it in more detail?
> > 3. The reward function $r$ is declared as a scalar in line 140 as "$r \in \mathbb{R}$", but in line 174, it is referenced as a function "$r \colon \mathcal{S}^L \times \mathcal{A} \to \mathbb{R}$". Although this is a minor point and I could guess what the authors really mean, this is not standard math writing and I would still suggest to make the definition consistent in the next revision.
> >
> > To sum up, I believe the paper can be strongly improved if these concerns about writing, experimental design and representation are addressed, but I will keep my score for now.

---

> ### Author Response · Authors · 2024-11-22
> **Response to Reviewer NeTy (Cont.)**
>
> 5. **Math**: We appreciate the reviewer’s observation. All suggestions are valid, and we have addressed them accordingly.
>
> 6. **Minor**: We appreciate the reviewer’s observation. We have changed the color of the explore map to improve the visualization.
>
> We thank again for the reviewer’s effort and humbly request that the reviewer consider raising their score if the above reply adequately addresses their concerns and articulates our contribution.
>
> **References:**
> [1] Haarnoja et al. "Soft actor-critic: Off-policy maximum entropy deep reinforcement learning with a stochastic actor." Proceedings of the 35th International Conference on Machine Learning (2018). [2] Garg et al. "Iq-learn: Inverse soft-q learning for imitation." Advances in Neural Information Processing Systems, 34 (2021). [3] Wulfmeier et al. "Imitating language via scalable inverse reinforcement learning." NeurIPS 2024 (2024).

---

> ### Author Response · Authors · 2024-11-29
>
> We thank Reviewer NeTy for their valuable time and effort in reviewing our work and responding to our rebuttal. We appreciate the additional suggestions provided and have made corresponding clarifications regarding the concerns mentioned in their reply:
>
> 1. **Experimental Details**
>    1.1 We have addressed Reviewer Qvd7's concerns regarding the experimental procedure in detail in our response to Qvd7 (https://openreview.net/forum?id=v7a4KET0Md&noteId=X5B2b73oi6).
>    1.2 **Number of random seeds:** Following the suggestions of all three reviewers, we have conducted experiments for all three datasets using the same number of random seeds and updated the results in the Nov 27 revised version of the paper. The updated results do not impact the conclusion of those experiments.
>
> 2. **Complexity and Scalability**
>    2.1 We acknowledge that solving a 50×50 gridworld environment (L=4) with the current implementation is not practical, although theoretically it can converge to at least a local optimum given sufficient time. As discussed in the scalability section of the appendix, larger environments are not the main focus of this study, as this work is centered on applications to neuroscience, particularly animal behavior studies. For scalability improvements, we propose exploring adaptations such as employing Soft Actor-Critic for the inner loops or IQ-Learn to convert IRL problem into a single-level inverse Q-learning problem as future work.
>    2.2 We acknowledge the need of providing empirical results on larger gridworld environments to help readers better understand the practical limits of the current implementation. We will conduct additional experiments on larger gridworlds, such as 25×25, to explore and report these empirical limits in the future revision.
>
> 3. **Minor Revisions**
>    3.1 We have removed the numbering of equations in Algorithm 1 as suggested by the reviewer.
>    3.2 It should be $G(\theta, \theta^k)$. We apologize for the oversight and have corrected it.
>    3.3 We have updated section 3.1 and 3.3 to ensure consistency in the $r$ function definition.
>
> We thank again for the Reviewer NeTy's valuable feedback and hope above clarifications address the reviewer’s concerns.

---

### Official Review · Reviewer_Qvd7 · 2024-11-02

**Soundness:** 2
**Presentation:** 3
**Contribution:** 2
**Rating:** 3
**Confidence:** 2

**Summary:**

This paper addresses the problem of inverse RL in a hidden-mode MDP, i.e. an MDP with an additional hidden-mode parameter that affects the reward. The authors propose an EM-style algorithm that learns both the reward function/policy and hidden mode in the given expert trajectories.
The authors then validate their approach on a synthetic gridworld task and go on to use it to model animal behavior in a rat maze, where the hidden mode represents the rat's current objective (i.e. get water, explore, go home).

**Strengths:**

* The method seems to work well and produces nicely interpretable result on the real world dataset.
 * The paper is well written and reads quite nicely.

**Weaknesses:**

* The literature review is limited. For example, a different interpretation for this problem setting would be a POMDP, for which previous IRL literature exists, e.g. [1]. Yet another interpretation of the mouse experiments would be a hierarchical RL setting, for example an Option setting in which options might be "get water" "explore" "go home". For this setting previous IRL method exist as well, e.g. [2]. I'm not sure if they are directly applicable to this paper's problem setting, but I think they might be applicable?
 * The method seems to be limited to relatively small discrete state and action sets, limiting it's general applicability.

**Experiments**
 * The most competitive baseline was labeled as "I-1" in plots, while the poorly performing baselines are labeled with their names. This might lead to it being confused as the author's contribution and is thus highly misleading and should definitelyl be corrected.
 * It is not clear how often each experiment was run, or how the box plots were created. How were outliers selected? Figure 3E eliminates the best result for baseline I-1.
 * It is also not clear what shaded areas in Figure 4B represent.
 * The MaxEnt baseline is missing in Figure 4? Why?

Minor points
 * L453 refers to an appendix which seems to be missing?
 * L202 $\xi$ is never defined in the paper?

References:
[1] Choi et al "Inverse Reinforcement Learning in Partially Observable Environments", JMLR, 2011
[2] Chen et al. "Option-Aware Adversarial Inverse Reinforcement Learning for Robotic Control", ICRA, 2023

**Questions:**

* Why is the problem modelled as a hidden-mode MDP rather than a POMDP or Hierarchical RL setting?

Overall I think this paper has potential, and if the issues with the experimental validation and related work discussed above are corrected I would be happy to increase my score.

---

> ### Author Response · Authors · 2024-11-22
> **Response to Reviewer Qvd7**
>
> We sincerely thank the reviewer Qvd7 for their detailed comments and analysis of our paper. We greatly appreciate the time and effort they dedicated to analyzing our work and providing constructive feedback. We are pleased that they found our paper to be interesting and well-written and recognized the potential of SWIRL.
>
> Below, we provide detailed responses to each concern raised under Weaknesses and Questions:
>
> 1. **POMDP**: It is indeed possible to formulate our problem within a POMDP framework; however, such a formulation is nontrivial and requires further investigation. One key challenge is that representing our problem as a POMDP introduces significant complexity into the inference procedure, which could be unnecessary for our objectives. The main focus of SWIRL is to perform IRL on animal behavior data with multiple switching reward functions, incorporating both decision-level and action-level dependencies. For this purpose, the HM-MDP framework, which is simpler than POMDP, is well-suited to address our problem. The existing Inverse POMDP literature [1] infers reward functions from expert demonstrations while assuming the Observation function $P(z|s, a)$ is already known—a condition that does not hold in our case. To the best of our knowledge, there is no existing Inverse POMDP method that can effectively address our problem.
>
> 2. **Hierarchical RL**: Hierarchical RL lacks a unified definition, as different Hierarchical RL frameworks introduce hierarchy in different ways. In SWIRL, HM-MDP provides a well-defined framework for our problem and can be viewed as one type of Hierarchical RL. Regarding H-AIRL [2], while it is not directly applicable to our setting, we acknowledge it has state-dependent option $z$ transition structure. However, its graphical model includes a future-option dependency, where the current action depends on the future option. This future-dependency differs from our history-dependency approach and is not biologically plausible in the context of our problem. As reviewer NeTy pointed out, another work [3] studies IRL with state-dependent hidden transitions, which is more aligned with our setup than H-AIRL. The graphical model of [3] can be viewed as S-1, making it a more relevant baseline for our framework. Consequently, we have cited [3] as a baseline work in our paper. Nonetheless, H-AIRL offers an insightful adversarial IRL perspective within the EM framework, which could inspire future development of SWIRL. We have cited [2] and included this discussion in the scalability and broader impact section of our paper.
>
> 3. **Scalability**: We have added a scalability and broader impact discussion in Appendix A.4. While scalability is not the main focus of this work, as typical behavior datasets in neuroscience have small or moderate state-action space and current SWIRL implementation can perform efficiently on them, we acknowledge the need for a more general and scalable implementation to address broader applications. In its current form, every step of the SWIRL inference procedure, except for the Soft-Q iteration, is compatible with large or continuous state-action spaces. However, the Soft-Q iteration is limited to discrete state-action spaces and can be slow with large state-action space. Nevertheless, for applications requiring scalability and compatibility with general state-action spaces, alternative methods can be adapted to replace the Soft Q iteration in the RL inner loop. For instance, Soft Actor-Critic [4]. Furthermore, a promising future direction is to reformulate the standard MaxEnt IRL $r$-$\pi$ bi-level optimization problem in SWIRL as a single-level inverse Q-learning problem, based on the IRL approach known as IQ-Learn [5]. This method has been successfully adapted to large language models training, demonstrating great scalability [6]. We also recognized the adversarial IRL perspective of H-AIRL and proposed it as a possible future direction for SWIRL.

---

> ### Author Response · Authors · 2024-11-22
> **Response to Reviewer Qvd7 (Cont.)**
>
> 4. **Presentation of experiments**:
>    - **4.1 Label baseline as I-1**: We articulated at the beginning of the results section that I-1 and S-1 can be viewed as baseline methods of Multi-intention IRL and locally consistent IRL (the [3] mentioned by reviewer NeTy). The notation I-1, S-1, I-2, and S-2 was chosen to highlight that SWIRL is a more general framework, with Multi-intention IRL and locally consistent IRL as specific cases within it. Researchers can leverage our SWIRL implementation to test various SWIRL models on animal behavior data, enabling the validation or rejection of hypotheses about history dependencies in animal behaviors. We apologize for not making this point clearer in the paper and will consider better ways to present I-1 and S-1 to avoid confusion in future revision.
>    - **4.2 How often each experiment is run**: We have included implementation details in Appendix B.1. Each experiment presented is the result of 10 runs initialized with different random seeds.
>    - **4.3 How boxed plot created**: We have included the relevant information in Appendix B.6 and referenced it in the captions of all box plots.
>    - **4.4 Shaded areas in Figure 4B**: The shaded area represents the total area that falls between one standard deviation above and below the mean. We have added this explanation to the captions of all plots with shaded areas to ensure clarity.
>    - **4.5 MaxEnt baseline is missing in Figure 4**: MaxEnt test LL is at the bottom right of the plot.
>
> 5. **Minor points**:
>    - **5.1 "L453 refers to an appendix which seems to be missing?"**: We appreciate the reviewer’s observation. We have added the corresponding figure in Appendix C.1 and corrected this reference.
>    - **5.2 "L202 $\xi$ is never defined in the paper?"**: It was defined in Section 3.2 and we apologize for not defining it explicitly in Algorithm 1. We have updated Algorithm 1 to include a clear definition of $\xi$.
>
> **Finally, we would like to highlight the key contributions of SWIRL in this work.** As an application-to-neuroscience study submitted to the neuroscience track, it proposed the SWIRL framework for IRL with switching reward functions that incorporated biologically feasible decision-level and action-level history dependencies, offering a valuable tool for analyzing animal behavior.
>
> SWIRL is the first model to incorporate action-level history dependency and we have demonstrated its significance in understanding animal behavior. For instance, in the Labyrinth experiment, the hidden mode segments in Fig. 3.F illustrate that S-1 fails to learn meaningful hidden mode segments, while S-2 succeeds. Additional investigations into S-3 ($L=3$) and S-4 ($L=4$) in Appendix B.4.2 show that higher $L$ values improve test log-likelihoods (LL), suggesting that animals in the labyrinth task rely on longer history for decision-making. In the spontaneous behavior experiment, while longer action-level dependency does not yield better results, the findings reinforce our claim that SWIRL serves as a versatile tool for hypothesis testing in behavioral datasets, enabling researchers to validate or challenge hypotheses regarding history dependencies. This adaptability underscores SWIRL’s potential to advance our understanding of complex behaviors.
>
> Additionally, we are the first to explore the connection between HM-MDPs and other popular dynamics models (e.g., ARHMMs, SLDS) used in animal behavior studies. By bridging these frameworks, we promote a more comprehensive understanding of behavior analysis tools and demonstrate how SWIRL can be adapted to experiments previously analyzed with these models.
>
> Moreover, our framework is easily extendable to large and continuous state and action spaces by replacing the Soft-Q iteration in the RL inner loop. This scalability positions SWIRL as not only a valuable tool for computational neuroscience but also a significant contribution to the broader machine learning community.
>
> We thank again for the reviewer’s effort and humbly request that the reviewer consider raising their score if the above reply adequately addresses their concerns and articulates our contribution.
>
> **References:**
> [1] Choi et al. "Inverse reinforcement learning in partially observable environments." JMLR (2011).
> [2] Chen et al. "Option-aware adversarial inverse reinforcement learning for robotic control." ICRA 2023 (2023).
> [3] Nguyen et al. "Inverse reinforcement learning with locally consistent reward functions." Advances in neural information processing systems, 28 (2015).
> [4] Haarnoja et al. "Soft actor-critic: Off-policy maximum entropy deep reinforcement learning with a stochastic actor." Proceedings of the 35th International Conference on Machine Learning (2018).
> [5] Garg et al. "Iq-learn: Inverse soft-q learning for imitation." Advances in Neural Information Processing Systems, 34 (2021).
> [6] Wulfmeier et al. "Imitating language via scalable inverse reinforcement learning." NeurIPS 2024 (2024).

---

> > ### Comment · Reviewer_Qvd7 · 2024-11-25
> >
> > I would like to thank the authors for the extensive rebuttal.
> > My conconcerns about related work have been mostly addressed, but I still have serious concerns about the evaluation methodology.
> >
> > Particularly, the new Appendix B.1 states that between 10–22 random seeds were run for each environment and the top 10 results by likelihood of the training data were then chosen.
> > The paper states that this is common practice in the EM literature, which I am not very familiar with, but it does strike me as non-standard. Additionally, adjusting the number of seeds for each experiment also seems questionable. At the very least, such details should be mentioned in the main text.
> > Futher, after varying the number of seeds per experiments, outliers are eliminated without any justification as to why this is done.
> >
> > Overall I still believe this paper has potential, but needs additional revisions and a more consistent experimental validation before acceptance.
> >
> > As the authors propose that their methodology is common in the EM literature I will reduce my confidence, but I am keeping the same score.

---

> ### Author Response · Authors · 2024-11-25
> **Response to Reviewer Qvd7**
>
> We thank Reviewer Qvd7 for their valuable time in reviewing our work and responding to our rebuttal. Below, we provide further clarifications regarding our experimental approach and its validation:
>
> **1. Selection of Top Results Based on Training LL**
> Selecting results by training LL is a necessary practice to ensure reliable outcomes in the EM algorithm, as widely adopted in neuroscience research. We have discussed the rationale (EM can be sensitive to local optima) in the appendix. For instance, Keypoint-Moseq employs EM to infer ARHMMs for segmenting animal behaviors and uses a likelihood-based metric to remove outlier models [1].
>
> **2. Experiment Validation: Going Beyond Existing Literature**
> We would like to emphasize that our experimental validation goes ABOVE and BEYOND previous works using EM for IRL with multiple reward functions. Below, we detail how our baselines and the paper suggested by Reviewer Qvd7 (H-AIRL [4]) conduct their experiments:
>
> - **Multi-Intention IRL [2]**
>   The paper only mentions conducting 5-fold cross-validation without specifying the number of runs or random seed selections. Upon reviewing their code on GitHub, we found that for each fold, they perform **10 runs** and select the **best test LL** as the test LL for that fold (**top 1 from 10 runs based on test LL**).
>
> - **Locally Consistent IRL [3]**
>   This work uses **3 random instances** (likely equivalent to 3 random seeds) to perform experiments and plot error bars (**3 seeds**).
>
> - **H-AIRL [4]**
>   Similarly, H-AIRL experiments use **3 random seeds** (**3 seeds**).
>
> - **SWIRL**
>   In contrast, we tested a **larger number of random seeds** (**top 10 seeds from 22 seeds** on Labyrinth) than these works to ensure reliability. For the spontaneous behavior dataset we only used 10 seeds and please see below response 3. for detailed explanation. Compared to previous work, we explicitly acknowledged this limitation of the EM algorithm (sensitive to local optima) in our analysis, which is a valuable contribution to the application of EM in IRL.
>
> **3. Adjusting number of seeds for each experiment**
> We have outlined the reasoning behind the number of seeds used for each experiment in detail below. We acknowledge that using varying numbers of seeds seems random and may lead to confusion. To address this, we will conduct additional runs to ensure consistency across all three experiments by adopting the same strategy: selecting the top 10 training LL results from 20 random seeds. (**UPDATE**: We have updated the results in the Nov 27 revision.)
> - **Gridworld:** We happened to conduct 1 more experiment when testing the swirl pipeline. As a result, we reported 11 random seeds. 10 random seeds instead should be totally fine. We didn't observe significantly lower training LL, which suggests that EM does not get stuck in bad local optima in this case.
> - **Labyrinth:**  On labyrinth experiment, we found that certain random seeds led to significantly lower training LL, which was in accordance with EM's limitation. To ensure we have enough meaningful results, we conducted more experiments with random seeds. Obviously we need more than 10 random seeds to report 10 meaningful results, but there is no specific reason for choosing the number of random seeds to be 22.
> - **Spontaneous Behavior:**  Spontaneous behavior dataset is a much smaller environment (only 9 states) than labyrinth and we didn't observe significantly lower training LL, which suggests that EM does not get stuck in bad local optima in this case. Therefore, we conclude that there is no need to do additional experiments and pick the top 10.
>
> **4. Mention in the Main Text**
> We acknowledge the importance of discussing these practices in the main text and will include them in a future revision due to time and page limits.
>
> **5. Outlier Selection in Seaborn Boxplots**
> We used the default outlier selection method in `seaborn.boxplot`. We would like to point out that outlier elimination in our plots does not affect their validity. For example, Reviewer Qvd7 noted that Figure 3E eliminates the best result for baseline I-1. However, including this outlier within the I-1 box would lead to only a minimal impact: I-1 would still outperform MaxEnt and remain inferior to I-2, S-1, and S-2. We will further clarify this point in future revisions.
>
> We hope these clarifications address the reviewer’s concerns. Thanks again for the feedback.
>
> **References:**
> [1] Weinreb et al. Keypoint-moseq: parsing behavior by linking point tracking to pose dynamics. Nature Methods (2024).
> [2] Zhu et al. Multi-intention inverse q-learning for interpretable behavior representation. Transactions on Machine Learning Research (2024).
> [3] Nguyen et al. Inverse reinforcement learning with locally consistent reward functions. Advances in neural information processing systems, 28 (2015).
> [4] Chen et al. "Option-aware adversarial inverse reinforcement learning for robotic control." ICRA 2023 (2023).

---

### Official Review · Reviewer_trZk · 2024-11-03

**Soundness:** 3
**Presentation:** 3
**Contribution:** 4
**Rating:** 6
**Confidence:** 3

**Summary:**

The paper introduces SWIRL (SWItching IRL), a novel framework that extends inverse reinforcement learning (IRL) by incorporating time-varying, history-dependent reward functions to model complex animal behaviour. Traditional IRL methods often assume a static reward function, limiting their ability to capture the shifting motivations and history-dependent decision-making observed in animals. SWIRL addresses this limitation by modelling long behavioural sequences as transitions between short-term decision-making processes, each governed by a unique reward function. It incorporates biologically plausible history dependency at both the decision level (transitions between decision-making processes depend on previous decisions and environmental context) and the action level (actions depend on the history of states within a decision-making process). The authors apply SWIRL to simulated data and two real-world animal behaviour datasets, demonstrating that it outperforms existing models lacking history dependency, both quantitatively and qualitatively. They also highlight connections between SWIRL and traditional autoregressive dynamics models, arguing that SWIRL offers a more generalized and principled approach to characterizing animal behaviour.

I think this is a very interesting and well-written paper. I gave a score of 6 but I am willing to reconsider this score if the authors can adequately address my concerns, particularly regarding the methodological details, hyperparameter selection, and theoretical analysis.

**Strengths:**

- **Originality/innovation**: The paper presents a novel extension to IRL by incorporating history-dependent reward functions, addressing a significant gap in modelling complex, naturalistic animal behaviours. This is the first IRL model to integrate both decision-level and action-level history dependency.
- **Quality/empirical validation**: The authors provide a thorough mathematical formulation of SWIRL, including detailed explanations of how history dependency is incorporated at different levels, and a clear demonstration of improvements over baseline methods. The choice of the authors to use both simulated and real-world datasets strengthens the validation of their approach.
- **Clarity**: The paper is generally well-written, with clear explanations of the concepts and methods. The connection drawn between SWIRL and traditional autoregressive dynamics models helps to contextualize the work within existing literature.
- **Significance**: The SWIRL framework offers a more accurate model of animal decision-making. Hence, I believe it has the potential to advance our understanding in neuroscience and behavioural science, opening up new ways for analysing long-term, complex behaviours driven by intrinsic motivations. Finally, the presented experiments are (in theory) reproducible with public datasets and publicly available code.

**Weaknesses:**

- **Methodology**: Some aspects of the implementation, such as hyperparameter selection and the specifics of the inference algorithm, are not fully detailed. Providing more information on these would enhance reproducibility and allow for better assessment of the method. There is limited analysis on hyperparameter sensitivity or discussion of how to choose the history length L in practice. In addition, the impact of number of hidden modes is not thoroughly explored. Finally, there is a lack of a theoretical analysis that would strengthen the paper, such as providing convergence guarantees, a discussion on optimality conditions.
- **Scalability**: The computational complexity of SWIRL, especially with history dependency and the EM algorithm, may pose challenges for large datasets. The paper would benefit from a discussion on scalability to larger state/action spaces and potential optimization strategies. Could also benefit from runtime comparisons with baseline methods.
- **Biological plausibility**: While the model is said to incorporate biologically plausible history dependency, the paper could provide more evidence or discussion on the biological validity of the specific mechanisms used.

**Questions:**

- **History length**: How does the choice of history length L in the action-level dependency affect the performance of SWIRL? Did you experiment with values beyond 1 and 2, and if so, what were the findings? What trade-offs did you observe between increased history length and computational complexity? What guidelines would you suggest for choosing L in practice?
- **Computational complexity**: Can you provide more details on the computational complexity of SWIRL compared to baseline models? How does it scale with the size of the dataset and the length of the history dependency? Could you provide specific runtime comparisons on the datasets used in the paper? Additionally, can you provide insights into the convergence properties of the EM algorithm?
- **Hyperparameter selection**: How were hyperparameters, such as the temperature parameter α in the soft-Q iteration, selected? Was any hyperparameter tuning performed, and if so, what criteria were used? Did you employ any cross-validation procedures to ensure the robustness of the results? How sensitive is the model to the choice of initial parameters?
- **Limitations**: Could you elaborate on any limitations of SWIRL in modelling certain types of animal behaviours? Are there situations where history dependency might not adequately capture the decision-making processes? How might SWIRL perform on behaviours with very long-term dependencies that extend beyond the history length L? Also, does this framework work in both discrete and continuous state/action spaces?
- **Robustness to noisy data**: How does this framework handle noisy or incomplete data, which are common in real-world animal behaviour datasets? Did you assess the robustness of the model under such conditions?
- **Minor:** line 83: In the experiment "section?" - or "In the Results section,"
- **Discussion suggestion:** Could this framework be used to provide evidence whether models of intrinsic reward (e.g. expected free energy or empowerment) are indeed able to capture animal behaviour?

---

> ### Author Response · Authors · 2024-11-22
> **Response to Reviewer trZk**
>
> We sincerely thank the reviewer trZk for their detailed and thoughtful comments on our paper. We greatly appreciate the time and effort they dedicated to analyzing our work and providing such constructive feedback. We are particularly encouraged by the reviewer’s recognition of SWIRL’s novelty and “potential to advance our understanding in neuroscience and behavioral science.”
>
> We will now respond to each concern raised in the Weaknesses and Questions.
>
> 1. **Methodology**: We apologize for not making this clearer in the paper. We have added implementation details and hyperparameters selection discussion to the Appendix B. Specifically, for questions the reviewer raised:
>
>    1.1 **Number of hidden modes**: We have included additional experiment results in Appendix B.4. For the labyrinth dataset, in Appendix B.4.1, we test the hidden modes from 2 to 5 and find that the test log-likelihood (LL) plateaus at 4 hidden modes. However, after taking a close look at the hidden segmentation and reward map, we find the 4 hidden modes result does not differ too much from 3 hidden modes: It mainly segments the explore mode into 2 explore modes with similar reward maps. As a result, we still show 3 hidden modes results in the main paper for simplicity. For the mouse spontaneous behavior experiment, in Appendix B.4.3, we show that we pick 5 hidden modes since the test LL plateaus at 5 hidden modes.
>
>    1.2 **Action-level history $L$**: In the main paper, we present results for $L=2$ (S-2) as it effectively demonstrates the benefits of incorporating non-Markovian action-level history dependency into SWIRL. S-2 provides meaningful hidden mode segments with interpretable reward maps, whereas $L=1$ (S-1) fails to capture these dynamics adequately. To further strengthen this claim, we included additional experimental results in Appendix B.4.2, where we evaluated history lengths ranging from $L=1$ to $L=4$ for the labyrinth dataset. Our findings indicate that $L=4$ (S-4) achieves the highest test log-likelihood (LL), suggesting that animal behavior in this dataset exhibits a longer non-Markovian dependency. This observation aligns with the partially observable nature of this 127-node labyrinth: The mouse may not know the whole environment, so it tends to rely on longer state history to inform its decision-making. Due to the mouse's limited knowledge of the entire environment, it likely relies on a longer history of prior states to guide its decision-making. For the mouse spontaneous behavior experiment, in Appendix B.4.3, we show that $L=1$ (S-1) is the optimal choice, as $L=2$ (S-2) consistently provides lower test log-likelihood (LL) compared to $L=1$ (S-1).
>
>    1.3 **Hyperparameter selection**: We have added the discussion of the selection of discount factor $\gamma$ and temperature $\alpha$ in Appendix B.3.
>
> 2. **Computational complexity and empirical runtime**:
>
>    2.1 **Complexity analysis**: We have added a complexity analysis in Appendix A.3. The key bottleneck is in the RL inner loop. Current Soft-Q iteration has a time complexity of $O(ZIS^{L}A^2)$, where $Z$ is number of hidden modes, $I$ is number of Soft-Q iterations, $S$ is number of states, $A$ is number of actions and $L$ is the history length. Although as we know the transition model $P(s'|s, a)$, the $S^L$ space can be reduced by removing impossible $(s, s')$ pairs, we recognize that longer $L$ in SWIRL does require a considerable amount of computing resources.
>
>    2.2 **Empirical runtime**: We have added an empirical runtime discussion in B.2. Empirically, SWIRL speed is not slow for typical animal behavior datasets. We implemented SWIRL efficiently with JAX, leveraging the features of vectorization and just-in-time compilation to improve our performance. Every SWIRL (S-2) experiment on all 3 datasets (simulated gridworld, labyrinth and dopamine) can finish within 15-30 minutes on a V100 GPU. For longer $L$, a S-4 experiment on labyrinth with 50 EM iterations take 2-3 hours to finish on a L40S GPU. We switch to a L40S GPU for $L=4$ due to the V100 GPU's insufficient memory capacity.

---

> ### Author Response · Authors · 2024-11-22
> **Response to Reviewer trZk (Cont.)**
>
> 3. **Scalability**. We have added a scalability and broader impact discussion in Appendix A.4. While scalability is not the main focus of this work, as typical behavior datasets in neuroscience have small or moderate state-action space and current SWIRL implementation can perform efficiently on them, we acknowledge the need for a more general and scalable implementation to address broader applications. In its current form, every step of the SWIRL inference procedure, except for the Soft-Q iteration, is compatible with large or continuous state-action spaces. However, the Soft-Q iteration is limited to discrete state-action spaces and can be slow with large state-action space. Nevertheless, for applications requiring scalability and compatibility with general state-action spaces, alternative methods can be adapted to replace the Soft Q iteration in the RL inner loop. For instance, Soft Actor-Critic [1]. Furthermore, a promising future direction is to reformulate the standard MaxEnt IRL $r$-$\pi$ bi-level optimization problem in SWIRL as a single-level inverse Q-learning problem, based on the IRL approach known as IQ-Learn [2]. This method has been successfully adapted to large language models training, demonstrating great scalability [3].
>
> 4. **Convergence analysis**: We have added a convergence analysis discussion in Appendix A.3. The convergence of the SWIRL inference procedure can be established based on the convergence guarantee of the EM algorithm, MaxEnt IRL and Soft-Q iteration.
>
> 5. **Limitations**:
>
>    5.1 SWIRL may face challenges in modeling complex animal behaviors where decision-making and hidden mode transitions are influenced by external environmental factors, internal alertness and physiological states. For instance, an animal foraging during daylight may exhibit different behaviors compared to nighttime, even when guided by the same internal food reward map. Another example involves the presence of predators. In an environment where predator locations are unknown, an animal's optimal foraging policy may not be heading directly to the food source. Instead, it might need to make decisions based on its alertness to the predators' presence. Additionally, physiological factors can further complicate behavior. For example, an animal intending to move right under a specific reward map might fail to execute the intended action due to poor physiological conditions, such as fatigue after prolonged activity or illness, resulting in its body to move straight instead. In such scenarios, SWIRL may struggle to accurately infer the true internal reward maps driving the animal's decisions.
>
>    5.2 As we discussed in the scalability section, the Soft-Q iteration in SWIRL is limited to discrete state-action spaces. However, we can replace the Soft-Q iteration by other methods such as Soft Actor-Critic and make SWIRL compatible with continuous state-action space.
>
> 6. **Robustness to noisy data**: We discuss additional experiment results in appendix B.5. In this gridworld experiment, we introduced random permutations to a percentage of the states and actions in the training data, ranging from 0% to 50%. As expected, performance decreased as the level of permutation increased. The model maintained high accuracy with less than 10% permutation. Between 10% and 30%, SWIRL still demonstrated stable performance, achieving reasonable reward correlations and hidden mode segmentation accuracy despite the noise. Permutation beyond 30% led to very noisy data and it became hard for the model to maintain high performance. These results suggest that SWIRL can tolerate moderate levels of noise or incomplete data, making it suitable for real-world animal behavior datasets where such challenges are common.

---

> ### Author Response · Authors · 2024-11-22
> **Response to Reviewer trZk (Cont.)**
>
> 7. **Biological plausibility**: We have added additional biological evidences about history dependencies in the introduction section and provided more discussion on non-Markovian labyrinth results in Appendix B.4.2.
>
> 8. **Minor**: We thank the reviewer for pointing it out. We have fixed it according to the reviewer’s suggestion.
>
> 9. **Discussion suggestion**: Yes, SWIRL could indeed be used to evaluate whether models of intrinsic reward, such as expected free energy or empowerment, effectively capture animal behavior. For instance, SWIRL can be applied to infer reward functions directly from animal behavior data. Separately, intrinsic rewards can be computed using various models, including expected free energy or empowerment. By comparing the reward functions inferred by SWIRL with those computed by intrinsic reward models, we can assess the validity of these models. An appropriate intrinsic reward model should yield reward functions that align with those inferred by SWIRL.
>
> We thank again for the reviewer’s effort and humbly request that the reviewer consider raising their score if the above reply adequately addresses their concerns.
>
> **References**:
> [1] Haarnoja et al. "Soft actor-critic: Off-policy maximum entropy deep reinforcement learning with a stochastic actor." Proceedings of the 35th International Conference on Machine Learning (2018).
> [2] Garg et al. "Iq-learn: Inverse soft-q learning for imitation." Advances in Neural Information Processing Systems, 34 (2021).
> [3] Wulfmeier et al. "Imitating language via scalable inverse reinforcement learning." NeurIPS 2024 (2024).

---

> > ### Comment · Reviewer_trZk · 2024-11-26
> >
> > I thank the authors for their detailed and thoughtful responses to my concerns. The additional materials provided in the appendices, particularly regarding hyperparameter selection, complexity analysis, and runtime comparisons, have helped clarify several technical aspects of SWIRL.
> >
> > However, after reviewing the complete discussion thread and the authors' responses to all reviewers, I believe maintaining my current score of 6 is appropriate (at least for now). While the authors have addressed many of my original concerns, some important methodological considerations have emerged:
> >
> > 1. The experimental methodology, particularly regarding seed selection and result filtering, would benefit from greater consistency across experiments. While selecting models based on training likelihood may be common in EM literature, using different numbers of random seeds across experiments (10-22) makes it more difficult to compare results fairly. A unified approach using the same number of seeds across all experiments would strengthen the validation.
> > 2. The scalability analysis provided is helpful, but the practical limitations with larger state spaces (as noted in the discussion with Reviewer NeTy) suggest that the method's applicability may be more constrained than initially apparent. While this doesn't diminish SWIRL's value for typical neuroscience datasets, it would be valuable to more explicitly acknowledge these limitations in the main text.
> > 3. The comparison with Nguyen et al. 2015 has been clarified, and I appreciate the authors' explanation of SWIRL's novel contributions, particularly regarding action-level dependency. However, I agree with other reviewers that this relationship could be more clearly presented in the main text.
> >
> > In conclusion, while I maintain my position that this paper makes valuable contributions to the field, particularly in neuroscience applications, my current score appropriately reflects both its strengths and limitations. I encourage the authors to address these methodological concerns in the final version, particularly by adopting a more consistent experimental validation approach.

---

> ### Author Response · Authors · 2024-11-29
>
> We thank Reviewer trZk for their valuable time and effort in reviewing our work and responding to our rebuttal. We have made the following clarifications regarding the concerns mentioned in their reply:
>
> 1. Following the suggestions of all three reviewers, we have conducted experiments for all three datasets using the same number of random seeds and updated the results in the Nov 27 revised version of the paper. The updated results do not impact the conclusion of those experiments.
>
> 2–3. We acknowledge the need to discuss those limits and relationships more in the main text. Given the time and page limit, we kept most of the updated content in the appendix but will definitely integrate some into the main text in future revisions to enhance clarity and accessibility.
>
> We thank again for the Reviewer trZk's valuable feedback and hope above clarifications address the reviewer’s concerns.

---

### Meta-Review · Area_Chair_RvnU · 2024-12-19

**Metareview:**

This paper presented an EM-based IRL algorithm for learning time-varying reward functions to model animal behavior. The main claim is that it is a novel approach to modeling time-varying reward functions and leads to better performance than alternative models for the task of modeling animal behavior. Experimental results back-up this second claim. There was debate about the novelty of the first claim.

Major concerns remained among the reviewers about the inconsistency of the experimental validation, relationship to prior IRL work that models time-varying reward functions (Nguyen et al. 2015), and the underinvestigated ability of the approach, which scales with $S^L$, to run in a larger setting (S is the cardinality of the state-space and $L$ is the number of history states), for example a 50x50 gridworld with L=4.

I concur with the reviewer's concerns here -- while the authors have attempted to address these concerns, the reviewers were not convinced that the paper was acceptable. I recommend that this paper be rejected in its current version.

**Additional Comments On Reviewer Discussion:**

During the discussion period, one of the major concerns was the nonstandard evaluation with a variety of counts of random seeds, as well as reporting the best results across the set of random seeds. The authors provided a final update in which the experiments were re-run with more random seeds, and the authors stated that the updated results do not impact the conclusion of those experiments. The reviewers did not comment on this update, and I think the current evaluation strategy and experimental results would require further review to determine if this concern is resolved. I concur with the reviewers that a comprehensive study of the computational efficiency of the approach is missing (the paper would be substantially improved if it were included).

Regarding the novelty claim, the authors did acknowledge the similarity to Nguyen et al. 2015 and pointed out a distinction, claiming that that method is like an ablated version of their current method. Additional experiments show that the approach compares similarly, although somewhat favorably, to the method from that paper.

---

### Decision · Program_Chairs · 2025-01-22

Reject